# Out of Many, One: Designing and Scaffolding Proteins at the Scale of the Structural Universe with Genie 2

## Abstract

Protein diffusion models have emerged as a promising approach for protein design. One such pioneering model is Genie, a method that asymmetrically represents protein structures during the forward and backward processes, using simple Gaussian noising for the former and expressive SE(3)-equivariant attention for the latter. In this work we introduce Genie 2, extending Genie to capture a larger and more diverse protein structure space through architectural innovations and massive data augmentation. Genie 2 adds motif scaffolding capabilities via a novel multi-motif framework that designs co-occurring motifs with unspecified inter-motif positions and orientations. This makes possible complex protein designs that engage multiple interaction partners and perform multiple functions. On both unconditional and conditional generation, Genie 2 achieves state-of-the-art performance, outperforming all known methods on key design metrics including designability, diversity, and novelty. Genie 2 also solves more motif scaffolding problems than other methods and does so with more unique and varied solutions. Taken together, these advances set a new standard for structure-based protein design.

## 1 Introduction

The design of proteins with novel structures and functions has emerged as a potent technology in therapeutic (Silva et al., 2019; Cao et al., 2020; Shanehsazzadeh et al., 2023) and industrial applications (Quijano-Rubio et al., 2021; Huddy et al., 2024). Generative AI has driven recent advances in protein design, most notably diffusion (Ho et al., 2020; Song et al., 2020) and flow matching (Lipman et al., 2022) models, as has the revolution in protein structure prediction sparked by AlphaFold 2 (Jumper et al., 2021). Proteins are one-dimensional polymers of amino acids ("sequences") that fold into three-dimensional shapes ("structures"). Generative protein models mirror this delineation, with most operating either in the sequence or structural domain. One rationale for sequence-based methods is that sequences are what ultimately get synthesized as functioning biomolecules, while structures require an additional structure-to-sequence map (inverse folding). Sequence-based models include EvoDiff (Alamdari et al., 2023), a discrete diffusion model that uses order-agnostic autoregressive diffusion with a ByteNet-style (Kalchbrenner et al., 2016) architecture for denoising. EvoDiff is a promising and complementary approach to structure-based design, currently the prevalent paradigm.

Structure-based methods (Trippe et al., 2022; Wu et al., 2024a; Lin and AlQuraishi, 2023; Ingraham et al., 2023; Yim et al., 2023b;a; Anand and Achim, 2022; Fu et al., 2024; Wang et al., 2024) focus on modeling structure space and typically employ separate inverse folding models such as ProteinMPNN (Dauparas et al., 2022) to propose plausible sequences given a generated structure. Their key rationale is that structure more closely associates with protein function than sequence. Among them, Genie performs diffusion on backbone atom coordinates and uses an SE(3)-equivariant denoiser to reason over a cloud of reference frames constructed from backbone coordinates. FrameDiff (Yim et al., 2023b) uses a diffusion process in SE(3) on backbone frames with an AlphaFold-inspired architecture for denoising. FrameFlow (Yim et al., 2023a) adopts the general architecture of FrameDiff but uses flow matching instead. Chroma (Ingraham et al., 2023) combines a correlated diffusion process that respects statistical properties of natural proteins with an efficient graph neural network. It also includes a separate design network that predicts sequences and side-chain atoms given a generated

backbone. More recently, Proteus (Wang et al., 2024) uses a similar diffusion process and architecture as FrameDiff but introduces graph triangle blocks that combine the expressiveness of triangle attention from AlphaFold 2 with faster runtimes by limiting attention to nearby residues.

The inter-connectedness of sequence and structure suggests that integrating their representations would advance protein design, particularly for conditional tasks that require pre-specified sequence or structural elements. Recent methods reflect this. One approach integrates sequence information as a condition of a structure-based diffusion process, as RFDiffusion (Watson et al., 2023) does when designing proteins with known sequence fragments. Another approach performs diffusion or flow matching in a joint sequence-structure space, as done by MultiFlow (Campbell et al., 2024) when it combines an SE(3) structural flow with a discrete sequence flow. There have also been attempts (Costa et al., 2023) at jointly encoding sequence and structure in a latent space and diffusing in this space; however, the approach remains nascent.

Whether encoded by sequence or structure, function is what is sought in protein design. Many functions, including interactions with small molecules and other proteins, are governed by few residues, or a *motif*. Achieving prescribed functions can thus often be distilled into designing a protein with a specific motif (*e.g.,* an enzyme active site (Wang et al., 2022) or antigen-binding site (Yang et al., 2021)), known as *motif scaffolding*. Diffusion models have shown success in this realm: Wu et al. (2024b) developed a sequential Monte Carlo sampler called Twisted Diffusion Sampler and applied it to FrameDiff to scaffold motifs while RFDiffusion and an updated FrameFlow (Yim et al., 2024) were explicitly trained on motif-conditioned tasks. Yet, current models cannot design proteins with multiple independent motifs, as they require inter-motif positions and orientations to be known *a priori*. Proteins often comprise independent functional sites, either as separate domains connected by a flexible linker or as one globular domain, such as an enzyme with multiple substrate binding sites or a scaffolding protein that engages multiple signaling ligands. The ability to design such proteins, which we term *multi-motif scaffolding*, would enable the development of new enzymes (Ebrahimi and Samanta, 2023), biosensors (Yang et al., 2021), and therapeutics that disrupt or enhance protein-protein interactions (Marchand et al., 2022). Concurrent with our work, Castro et al. (2024) employed an established non-diffusion model, $RF_{joint2}$, to inpaint an immunogen containing three distinct epitopes. This approach appears promising but has yet to be systematically benchmarked.

In this work, we extend Genie to support single- and multi-motif scaffolding. We also improve the core Genie model through architectural modifications and enhancements to its training data and process. The resulting Genie 2 better captures protein structure space. When compared to existing models, Genie 2 sets state-of-the-art results in designability, diversity, and novelty. In addition, Genie 2 surpasses RFDiffusion on motif scaffolding tasks, both in the number of solved problems and the diversity of designs. We also curate a benchmark set comprising 6 multi-motif scaffolding problems from the literature and show that Genie 2 can propose complex designs incorporating multiple functional motifs, a challenge unaddressed by existing protein diffusion models.

## 2 PREVIOUS GENIE MODEL

**Diffusion with asymmetric protein representations** In contrast to all other SE(3)-equivariant diffusion models for protein generation, which use unified representations for the forward and backward diffusion processes, Genie represents proteins as point clouds of $C_\alpha$ atoms in the forward process and as clouds of reference frames in the reverse process. Let $\mathbf{x} = [\mathbf{x}^1, \mathbf{x}^2, \cdots, \mathbf{x}^N]$ be a sequence of $C_\alpha$ coordinates of length $N$. Given a sample $\mathbf{x}_0$ from the unknown protein structure distribution, Genie's forward process gradually adds isotropic Gaussian noise through a cosine variance schedule $\beta = [\beta_1, \beta_2, \cdots, \beta_T]$, where $T$ is the total number of diffusion steps (set to 1,000).

$$q(\mathbf{x}_t|\mathbf{x}_{t-1}) = \mathcal{N}(\mathbf{x}_t|\sqrt{1-\beta_t}\mathbf{x}_{t-1}, \beta_t\mathbf{I}) \qquad (1)$$

By reparameterization, we have

$$q(\mathbf{x}_t|\mathbf{x}_0) = \mathcal{N}(\mathbf{x}_t|\sqrt{\bar{\alpha}_t}\mathbf{x}_0, (1-\bar{\alpha}_t)\mathbf{I}) \quad \text{where} \quad \bar{\alpha}_t = \prod_{s=1}^{t}\alpha_s \quad \text{and} \quad \alpha_t = 1-\beta_t \qquad (2)$$

Since the isotropic Gaussian noise added at each diffusion step is small, the corresponding reverse process could be approximated with a Gaussian distribution:

$$p(\mathbf{x}_{t-1}|\mathbf{x}_t) = \mathcal{N}(\mathbf{x}_{t-1}|\mu_\theta(\mathbf{x}_t, t), \mathbf{\Sigma}_\theta(\mathbf{x}_t, t)\mathbf{I}) \qquad (3)$$

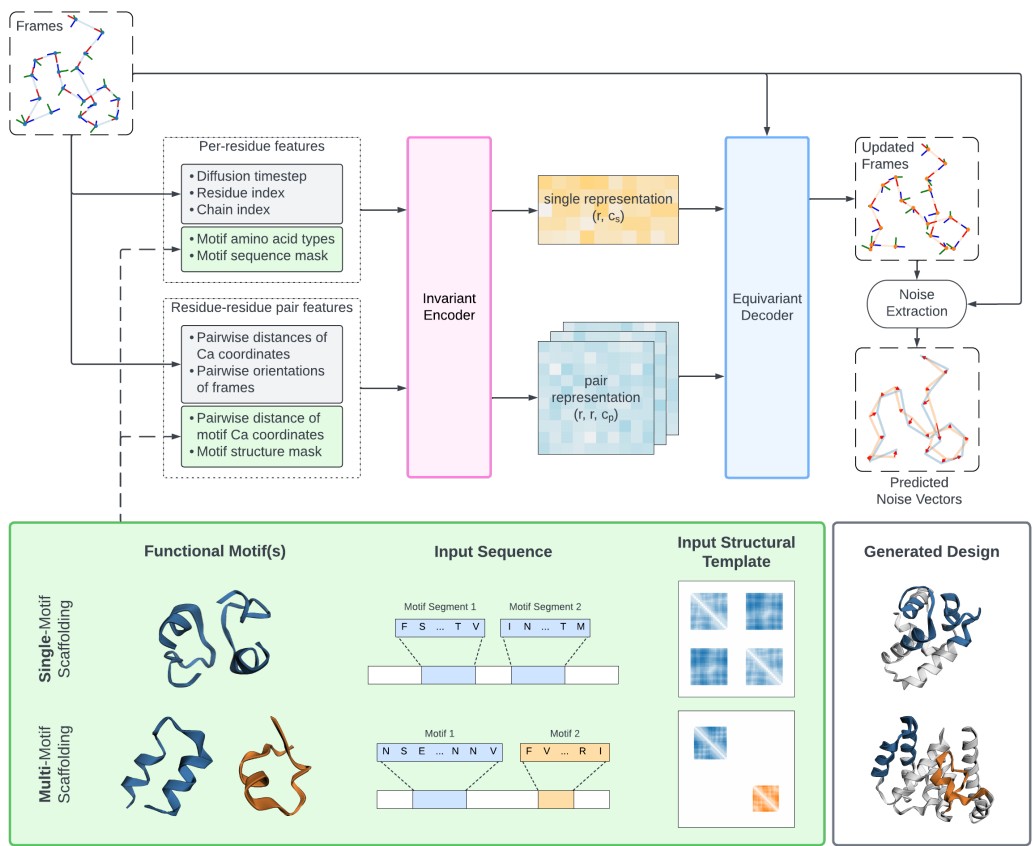

Figure 1: Genie 2 architecture (top), which extends Genie to enable scaffolding on (multiple) motifs. It consists of an SE(3)-invariant encoder that transforms input features into single residue and pair residue-residue representations, and an SE(3)-equivariant decoder that updates frames based on single representations, pair representations, and input reference frames. Example inputs to the model for single- and multi-motif scaffolding problems are shown (bottom-left green box), along with the corresponding generated designs (bottom-right box). In single motif scaffolding (top row), the motif may be contiguous or non-contiguous but all inter-residue positions and orientations are defined. In multi-motif scaffolding (bottom row), inter-motif geometry is left unspecified. For input sequences, white boxes denote masked out regions corresponding to the scaffold.

where

$$\mu_\theta(\mathbf{x}_t, t) = \frac{1}{\sqrt{\alpha_t}} \left( \mathbf{x}_t - \frac{\beta_t}{\sqrt{1 - \bar{\alpha}_t}} \epsilon_\theta(F(\mathbf{x}_t), t) \right) \qquad \mathbf{\Sigma}_\theta(\mathbf{x}_t, t) = \gamma^2 \cdot \beta_t$$

$F(\cdot)$ is the Frenet-Serret frame construction process based on a sequence of coordinates (Appendix A.1), and $\gamma \in [0, 1]$ controls the scale of injected noise in the reverse process (analogous to sampling temperature).

**SE(3)-equivariant denoiser** The core of Genie is its SE(3)-equivariant denoiser $\epsilon_\theta(F(\mathbf{x}_t), t)$, which reasons over reference frames to predict the noise injected during the forward process. Figure 1 summarizes Genie's architecture. The denoiser consists of an SE(3)-invariant encoder, which transforms individual residue and residue-residue pair features into single and pair representations, and an SE(3)-equivariant decoder, which uses Invariant Point Attention (Jumper et al., 2021) to update single representations that are in turn used to update input reference frames. Final noise vectors are computed as the displacement between the translation component of the updated frames and that of the input frames. For more details refer to Lin and AlQuraishi (2023).

## 3 METHODS

In this work we extend Genie's architecture and training procedure to enable motif scaffolding. We also substantially improve the core unconditional model through data augmentation and scaling.

**Motif representation for conditional generation**    Genie's architecture naturally permits integration of conditional sequence and structure information into the diffusion process. We do so by encoding the residues of each motif as one-hot vectors and concatenating these encodings to the single residue features. We encode the structure of each motif using the pairwise distance matrix of its $C_\alpha$ atoms. This representation is SE(3)-invariant as it does not encode the absolute position and orientation of the motif(s), and is unlike the motif conditioning procedures of other methods (*e.g.,* RFDiffusion and FrameFlow), which fix motif coordinates and are thus sensitive to initial placement(s).

Our approach sidesteps a challenge in multi-motif scaffolding, where the design objective leaves the relative positions and orientations of motifs unspecified. By representing motif structures using pairwise distance matrices that specify intra-motif but not inter-motif distances, Genie 2 learns to satisfy the constraints of each motif while generating self-consistent configurations of inter-motif geometries. Figure 1 illustrates the types of (multi-)motif templates that can be specified. Note that even in single motif scaffolding, a motif may be non-contiguous by comprising multiple segments. What differentiates single and multi-motif scaffolding is that inter-segment geometric relationships are specified while inter-motif relationships are not. Genie 2's formulation does require specifying sequence length separations between motifs, either by fixing them or sampling from a distribution.

**Training**    Genie 2 is trained in a purely conditional manner with every training example constituting a (single) motif scaffolding task. Tasks are constructed by first sampling structures from our training dataset to serve as ground truths. A target motif is then constructed for each structure by sampling $N_s$ segments totaling $N_r$ residues, where $N_s \sim \mathcal{U}(1,4)$, $N_r \sim \mathcal{U}(\lfloor 0.05N \rfloor, \lceil 0.5N \rceil)$, and $N$ is again the length of the protein. The starting positions and lengths of motif segments are randomly chosen subject to the number of motif residues totalling $N_r$. Algorithm 1 describes the task sampling procedure in more detail. We initially experimented with training on varying ratios of conditional and unconditional tasks but found that higher proportions of conditional tasks generally yielded better performance on both types of tasks, and thus switched to purely conditional training. We include an analysis of this behavior in Appendix B.1. Due to computational constraints, we limit sequence length to 256 during training; however, Genie 2 is capable of generating proteins longer than 256 residues. In addition, we do not train on multi-motif scaffolding as our input representation permits under-specification of geometric relationships as an inference-time choice. Genie 2's performance on multi-motif scaffolding thus represents out-of-distribution generative generalization.

---

**Algorithm 1** Motif construction for conditional training task

---

**Require:** Sampled structure $\mathbf{x}$, a sequence of $C_\alpha$ coordinates of length $N$

$\quad N_s \sim \mathcal{U}(1,4)$                                        ▷ Number of segments in the motif

$\quad N_r \sim \mathcal{U}(\lfloor 0.05N \rfloor, \lceil 0.5N \rceil)$                    ▷ Number of residues in the motif

$\quad B \leftarrow [0, b_1, b_2, \cdots, b_{N_s-1}, N_r] \quad$ where $b_1, b_2, \cdots, b_{N_s-1}$

$\quad\quad$ are randomly sampled from $\{1, 2, \cdots, N_r - 1\}$

$\quad\quad$ without replacement and sorted in ascending order.

$\quad L \leftarrow [l_1, l_2, \cdots, l_{N_s}] \quad$ where $L_i = B_i - B_{i-1}$          ▷ Split motif residues into segments

$\quad \mathbf{M} = \text{Flatten}(\text{Permute}([S_1, S_2, \cdots, S_{N-N_r}, M_1, M_2, \cdots, M_{N_s}])) \quad$ where

$\quad\quad S_i = [0]$ for $i \in [1, N - N_r]$                ▷ Represents a scaffold residue

$\quad\quad M_j = [1, 1, \cdots, 1]$ where $|M_j| = l_j$ for $j \in [1, N_s]$      ▷ Represents a motif segment

**return M**     where for $i \in [1, N]$              ▷ Represents a motif sequence mask

$\quad\quad \mathbf{M}[i] = 1$ indicates that residue $i$ is a motif residue

$\quad\quad \mathbf{M}[i] = 0$ indicates that residue $i$ is a scaffold residue

---

**Data augmentation**    Diffusion models require large datasets to robustly capture complex distributions. Generative protein models have thus far relied on training on experimentally determined

protein structures from the Protein Data Bank (PDB) (Berman et al., 2002; Burley et al., 2023). Despite the enormous experimental efforts that have gone into assembling the PDB, its size remains limited to ~20,000 proteins of relevant lengths. With the development of highly accurate protein structure prediction, we hypothesized that augmenting Genie training with confidently predicted protein structures could boost its performance by expanding the space of observed folds beyond those present in the PDB. Consequently, we train Genie 2 using the AlphaFold database (AFDB) (Varadi et al., 2022), which consists of approximately 214M AlphaFold 2 predictions spanning nearly the entirety of UniProt (Consortium, 2023). As AFDB is highly structurally redundant, we use a subsampled version (Barrio-Hernandez et al., 2023) that applies FoldSeek (Van Kempen et al., 2024) to cluster entries based on structural similarity. We start with all cluster representatives from the FoldSeek-clustered database and then filter them using a pLDDT threshold of >80, to enrich for highly confident predictions, and a maximum sequence length of 256. This results in 588,570 structures. For comparison, we also train a version of Genie 2 on the PDB and observe that the performance of PDB-trained Genie 2 falls behind that of AFDB-trained Genie 2. Further details and results on PDB-trained Genie 2 are included in Appendix B.3.

**Loss function**  We minimize the loss function below, which computes the mean squared error between predicted and ground truth noise:

$$L(\theta) = \mathbb{E}_{t,x_0,\epsilon} \left[ \frac{1}{N} \sum_{i=1}^{N} \left\| \epsilon_t^i - \epsilon_\theta^i(F(x_t), t) \right\|^2 \right] \tag{4}$$

$$= \mathbb{E}_{t,x_0,\epsilon} \left[ \frac{1}{|\mathcal{M}| + |\mathcal{S}|} \left( \sum_{i \in \mathcal{M}} \left\| \epsilon_t^i - \epsilon_\theta^i(F(x_t), t) \right\|^2 + \sum_{i \in \mathcal{S}} \left\| \epsilon_t^i - \epsilon_\theta^i(F(x_t), t) \right\|^2 \right) \right] \tag{5}$$

where $\mathcal{M}$ and $\mathcal{S}$ are the set of motif and scaffold residue indices, respectively. Under this construction, motifs are enforced as a soft constraint, ensuring that the model is responsive to motif specifications while also designing the protein as a whole.

## 4 UNCONDITIONAL PROTEIN GENERATION

To systematically assess Genie 2 and competing methods on unconditional protein generation, we conduct two sets of analyses. First, we assess methods without accounting for length while restricting the longest designed protein to 256 residues (Section 4.2). This reflects Genie 2's in-distribution generative power since it is trained on proteins up to 256 residues long. Second, we assess methods in a length-specific manner up to 500 residues (Section 4.3) to quantify Genie 2's out-of-distribution generative capabilities. In both analyses, we rely on the evaluation metrics described in Section 4.1. We compare Genie 2 to Chroma, FrameFlow, Proteus and RFDiffusion. The latter is widely perceived as the current state-of-the-art protein design model and has been extensively validated. We note that while Chroma contains a built-in sequence design network, we find it to underperform ProteinMPNN and so exclude it; instead we adopt the same evaluation pipeline across all methods.

### 4.1 EVALUATION METRICS

**Designability**  A structure that can be plausibly realized by some protein sequence is one that is designable. To determine if a structure is designable we employ a commonly used pipeline (Trippe et al., 2022) that computes *in silico* self-consistency between generated and predicted structures. First, a generated structure is fed into an inverse folding model (ProteinMPNN (Dauparas et al., 2022)) to produce 8 plausible sequences for the design. Next, structures of proposed sequences are predicted (using ESMFold (Lin et al., 2022)) and the consistency of predicted structures with respect to the original generated structure is assessed using a structure similarity metric (TM-score (Zhang and Skolnick, 2004; Xu and Zhang, 2010)). Using this pipeline, we consider a generated structure to be designable if it is within 2Å RMSD of the most similar predicted structure (scRMSD $\leq 2$) and the structure is confidently predicted (mean pLDDT $\geq 70$). Over a set, "designability" quantifies the fraction of designable structures within it. We note that designability alone can be misleading because it does not account for structural diversity–for example, a model that has mode-collapsed into a single designable structure achieves perfect designability.

Table 1: Unconditional generative performance of structure-based diffusion models.

| METHOD | DESIGNABILITY | DIVERSITY | $F_1$ | PDB NOVELTY | AFDB NOVELTY |
|---|---|---|---|---|---|
| CHROMA | 0.70 | 0.51 | 0.59 | 0.13 | 0.04 |
| PROTEUS | 0.90 | 0.30 | 0.45 | 0.04 | 0 |
| RFDIFFUSION | **0.96** | 0.63 | 0.76 | 0.26 | 0.14 |
| GENIE 2 | **0.96** | **0.91** | **0.93** | **0.41** | **0.21** |

**Diversity** Complementing designability is the diversity of a generated protein set. To quantify diversity we start by hierarchically clustering (with single linkage) the set of *designable* structures. We exclude non-designable structures as we do not expect them to be realizable and including them would thus inflate diversity. As sequence lengths vary within a set of structures, we use TMAlign (Zhang and Skolnick, 2005) to compute the pairwise TM scores of all structures and use a threshold of 0.6 as cutoff (thus any pair of structures across clusters would have a TM score of at most 0.6). We then compute "diversity" as the fraction of distinct designable clusters within the set. As diversity already enforces designability of generated clusters, we find that it better reflects model capability than designability. Note that diversity depends on the number of samples generated, and tends to 0 as sample size increases. In all our experiments we use a fixed sample size for even comparisons.

**F1 score** Following Lin and AlQuraishi (2023), we compute the harmonic mean between designability ($p_{\text{structures}}$) and diversity ($p_{\text{clusters}}$) as follows:

$$F_\beta = (1 + \beta^2) \cdot \frac{p_{\text{structures}} \cdot p_{\text{clusters}}}{\beta^2 \cdot p_{\text{structures}} + p_{\text{clusters}}} \tag{6}$$

where $\beta \in \mathbb{R}^+$ controls the relative weighting of designability and diversity. We set $\beta = 1$ and report the metric as F1 score.

**Novelty** Beyond designability and diversity, we also quantify the novelty of generated structures with respect to reference datasets and, by extension, the known structural universe. To compute the novelty of a generated structure we again employ TM-score as our structure similarity metric and use TMAlign to compute the TM scores between a generated structure and all structures in a reference dataset. We consider a generated structure to be novel if it is designable and its TM-score to any reference structure is at most 0.5. Similar to our diversity calculations, we apply hierarchical clustering (with single linkage and a TM-score threshold of 0.6) to the set of novel structures and define "novelty" to be the fraction of distinct novel clusters within a set of generated structures. We measure novelty with respect to both the PDB and Foldseek-clustered AFDB datasets (the latter being our training dataset) and term these measures "PDB novelty" and "AFDB novelty", respectively.

## 4.2 IN-DISTRIBUTION PERFORMANCE ANALYSIS

We assess Genie 2, Chroma, Proteus and RFDiffusion by generating 5 structures of every length ranging from 50 to 256 residues (1,035 structures in total). We omit FrameFlow here since it is trained using a maximum sequence length of 128, but include direct comparisons with FrameFlow in Section 4.3. Table 1 summarizes the performance of all methods on our key metrics. Relative to Proteus and RFDiffusion, Genie 2 achieves comparable designability and much higher diversity and novelty. This suggests that as a core unconditional model, Genie 2 best captures foldable protein structure space, and may thus serve as a superior engine for downstream sampling-based protein design tasks (Wu et al., 2024b; Didi et al., 2023).

In Figure 2A, we visualize the secondary structure distribution of generated proteins. While all methods yield a wide range of secondary structure elements, the resulting distributions are biased (relative to AFDB), with beta strand-containing structures (top left of distribution) and loop elements (bottom left) being generally underrepresented. There are multiple possible reasons for this bias. First, the high frequency of helices in the training dataset leads to models that favor generating helical structures. Second, alpha helices are likely easier to generate than beta sheets as they involve largely local interactions while sheets may involve long-range interactions. Third, we assess Genie 2 at a low sampling noise scale as it yields better results, but this may shift the model from its learned distribution. We test this hypothesis by visualizing the distribution of secondary structures generated

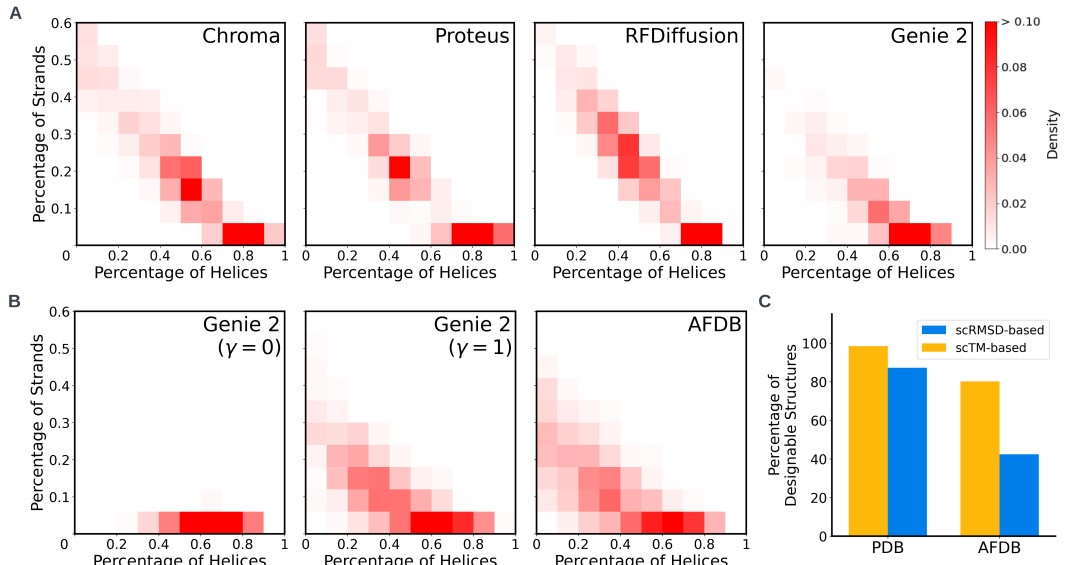

Figure 2: Visualizations of in-distribution performance on unconditional generation. **(A)** Secondary structure distributions of proteins generated by Chroma, Proteus, RFDiffusion and Genie 2. **(B)** Secondary structure distributions of proteins generated by Genie 2 when sampling noise scale ($\gamma$ in equation (3)) is set to 0 and 1. For reference, we also include the secondary structure distribution of 1,000 structures randomly drawn from AFDB (far right). **(C)** Self-consistency results on 1,000 randomly chosen structures from the PDB and clustered AFDB datasets.

by Genie 2 under a normal noise scale ($\gamma = 1$) in Figure 2B. We observe that the resulting distribution is in fact consistent with that of the clustered AFDB dataset.

This raises the question of whether low sampling noise scale is necessary or, alternatively, why it helps improve Genie 2's performance. To investigate this we ran our self-consistency pipeline on 1,000 randomly chosen structures from the clustered AFDB dataset. We found that only 42.4% of these structures are designable. When our designability criteria is relaxed to scTM $\geq 0.5$ and pLDDT $\geq 70$, this number rises to 80.2% (Figure 2C). For comparison, 87.2% of PDB structures are considered designable by the original criteria. Since we use AFDB for training, this might explain why designability is low at normal noise scale, while lower noise scale leads to higher fidelity. In effect, sampling at a low noise scale enables Genie 2 to leverage a larger structural dataset (with lower fidelity) during training while maintaining high fidelity at sampling time. We also note that ProteinMPNN was only trained on the PDB and thus it is possible that the apparent discrepancy in designability between the PDB and AFDB is due to a bias in ProteinMPNN towards the PDB.

### 4.3 Length-based performance analysis

We next assess generative performance in a length-dependent manner. For a subset of sequence lengths ranging from 50 to 500 residues, we generate 100 structures and assess them using our design metrics. Figure 3A shows the scRMSD distribution across sequence lengths while Figures 3B and 3C plot designability and diversity as a function of sequence length, respectively. At nearly all assessed lengths, Genie 2 has comparable designability to RFDiffusion but higher diversity. For short proteins (<200 residues), Genie 2 exhibits considerably higher diversity (doubling that of RFDiffusion at 100 residues), which is noteworthy as shorter lengths constitute smaller design spaces. While Proteus achieves higher designability than Genie 2 and RFDiffusion for longer proteins, its diversity (number of unique designable structures) is comparable to that of Genie 2 and RFDiffusion for longer proteins (>400 residues) and substantially worse for shorter proteins (<300 residues).

Generative models generally struggle to create larger proteins due to their increased complexity. As sequence length increases, designability decreases and in turn so does diversity, likely because diversity depends on the number of designable proteins. Larger protein lengths should in principle permit greater diversity but they are harder to generate. Nonetheless, despite having been trained on

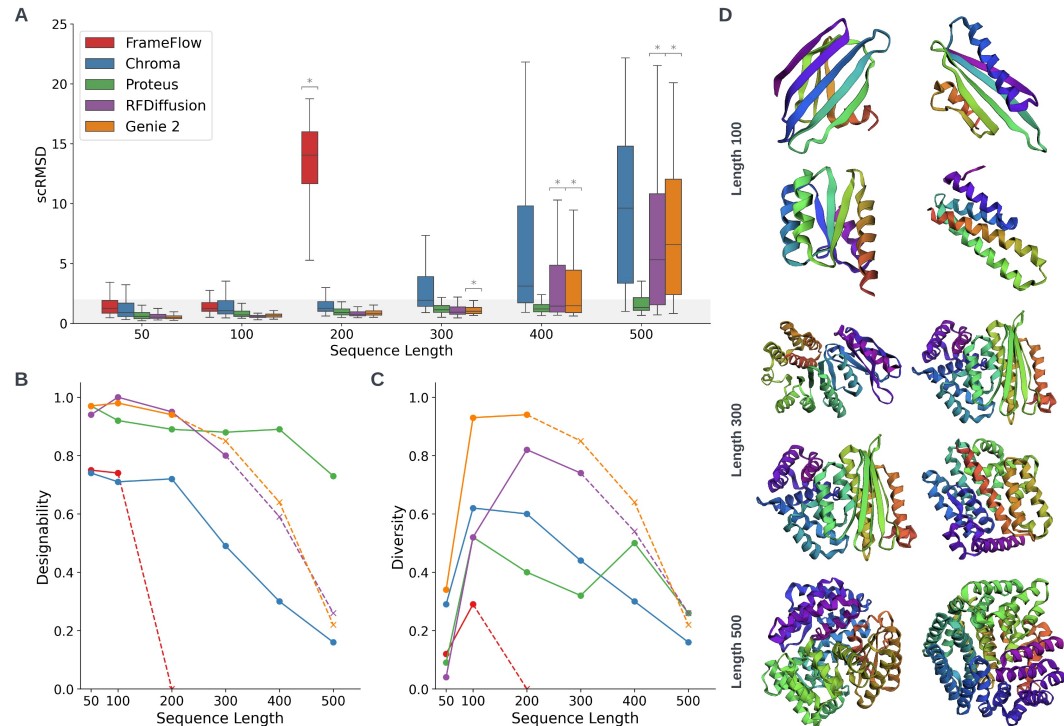

Figure 3: Length-based assessment of methods. For each method/sequence length combination, we generate 100 structures. **(A)** Box-and-whisker plots of scRMSDs between generated structures and their most similar ESMFold-predicted structures. Asterisks (*) indicate that sequence lengths exceed the maximum seen during training. **(B-C)** Plots of designability (B) and diversity (C) as a function of sequence length, with the same color scheme as (A). Out-of-distribution generation performance is represented with crosses and dashed lines. **(D)** Representative structures generated by Genie 2.

monomers of at most 256 residues, Genie 2 can generate 500-residue structures with comparable or better performance than competing methods. For reference, RFDiffusion uses a crop size of 384 during training while Chroma and Proteus train on even larger proteins (>500 residues) owing to their efficient graph neural networks. Figure 3D shows examples of Genie 2 designed structures.

## 5 MOTIF SCAFFOLDING

In this section we assess Genie 2 on single motif scaffolding and compare it to RFDiffusion and FrameFlow-amortized (Yim et al., 2024) (henceforth "FrameFlow") on a common set of design tasks. We also assess Genie 2 on multi-motif scaffolding using a suite of 6 multi-motif tasks that we curated.

### 5.1 EVALUATION METRICS

Motif scaffolding tasks consist of sequence and structure constraints on motif(s) plus length (min/max) constraints on the scaffolds and protein. To solve a given task, we first sample a constraint-satisfying length for each scaffold segment while ensuring that total protein length is also within specifications. These lengths, along with the sequence and structure of motif(s), are passed as conditions to Genie 2. We quantify success using the criteria of RFDiffusion, which requires that generated structures achieve scRMSD $\leq$ 2Å, pLDDT $\geq$ 70, and pAE $\leq$ 5 to be considered designable, and for designed motif(s) to have backbone RMSD $\leq$ 1Å with respect to each motif to be considered constraint-satisfying.

While most previous studies, including RFDiffusion, use success rate as the evaluation metric, we find that this tends to inflate performance, as it is possible to achieve high success rates by repeatedly generating only one or a few successful designs, *i.e.,* while suffering from mode collapse. Instead, and similar to Yim et al. (2024), we cluster successful designs based on structure similarity and report

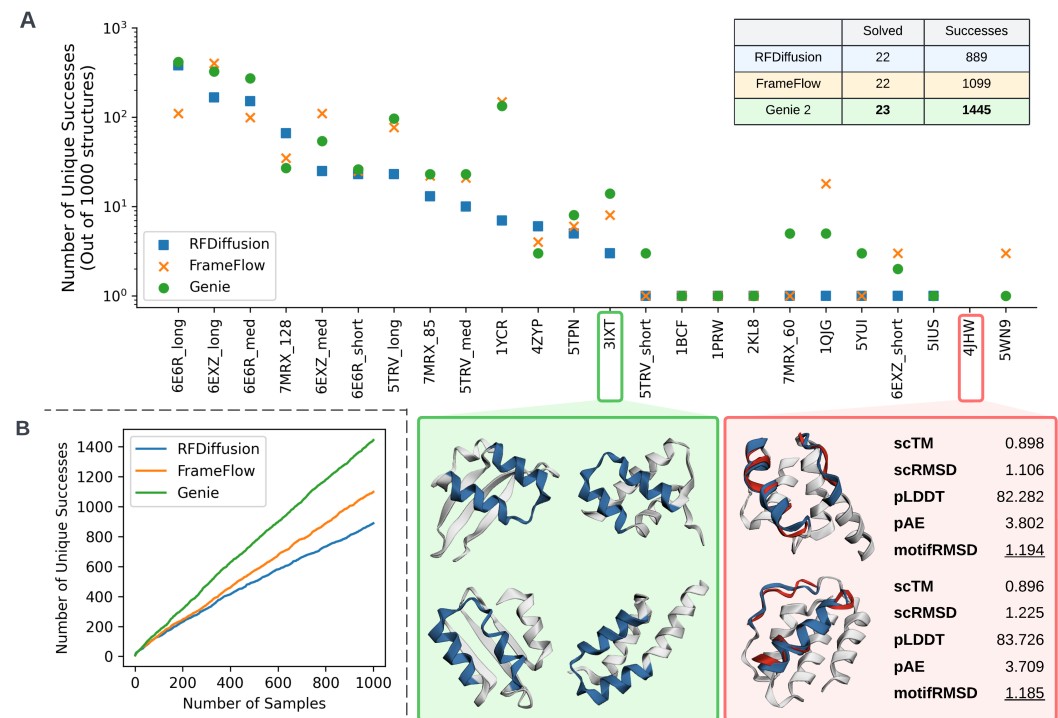

Figure 4: Comparison of Genie 2, RFDiffusion, and FrameFlow on single-motif scaffolding. **(A)** Performance of models across 24 single-motif scaffolding tasks. Summary statistics are shown in table (right). Example Genie 2 designs are shown (bottom) for successful task 3IXT (green) as well as failed task 4JHW (red). Scaffolds (white), motifs (blue), and unsatisfied sought motifs (red) are overlaid. **(B)** Plot of number of unique successes as a function of sample size.

the number of unique successes. This approach better balances designability with diversity when assessing motif scaffolding performance. We use hierarchical clustering with single linkage and a TM-score threshold of 0.6. For each motif scaffolding problem, we sample 1,000 structures.

## 5.2 SINGLE-MOTIF SCAFFOLDING

For evaluation we use an existing benchmark (Watson et al., 2023) comprising 25 tasks curated from 6 recent publications. We exclude one task, 6VW1, as its motif consists of segments from multiple protein chains, a requirement not supported by Genie 2. Figure 4A summarizes the performance of FrameFlow, RFDiffusion, and Genie 2. Relative to other methods, Genie 2 yields a greater number of unique designs and solves more motif scaffolding tasks. We show examples of successful designs in Figure 4A and Appendix F.5. We observe that the gap in number of unique successes between Genie 2 and competing methods widens as sample size increases (Figure 4B), suggesting Genie 2 captures a larger and more diverse structure space than other methods. Genie 2 does fail on one problem, 4JHW, that RFDiffusion and FrameFlow also fail on, which involves scaffolding the RSV F-protein site-0. To better understand this failure case, we visualize the two closest designs (red box in Figure 4) and observe that while Genie 2 yields designable structures it does not satisfy the motif constraints.

Analyzing individual motif scaffolding tasks, we observe that Genie 2 generates more unique designs for 12 problems while FrameFlow generates more unique designs for 8 problems, suggesting that the two methods complement each other well.

In addition to the above backbone-based analysis, we note that the validity of side-chain atom configurations can be essential for successful motif scaffolding. To address this question we perform an analysis with one additional constraint, requiring all motif atoms to be within 2Å of the target motif. We observe similar trends to the above, which we summarize in Appendix F.3.

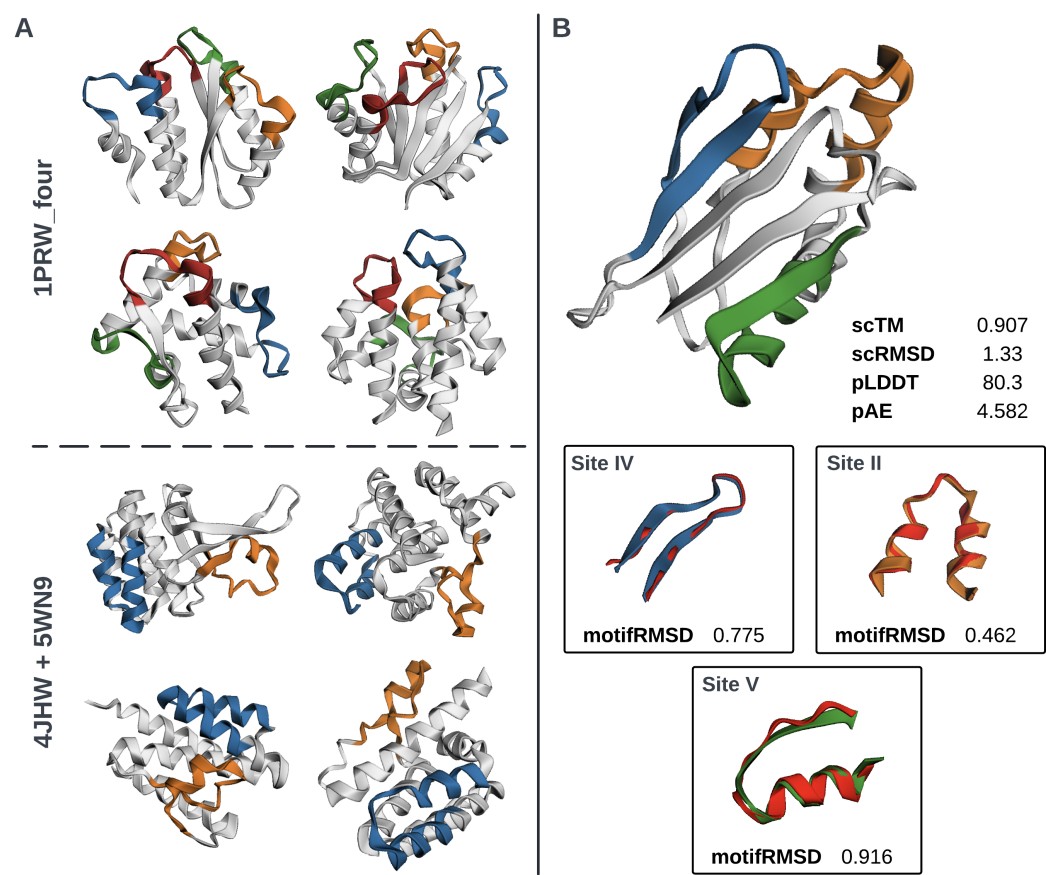

Figure 5: Multi-motif scaffolding. **(A)** Successful designs for task 1PRW_four (scaffolding with four $Ca^{2+}$ ion binding sites) and 4JHW+5WN9 (scaffolding with RSV-F site II epitope and RSV-G 2D10 epitope). Scaffolds are in grey and different motifs are colored distinctly. **(B)** (Top) Successful design of a multi-epitope immunogen. (Bottom) Individual epitope designs superposed over targets (red).

### 5.3 MULTI-MOTIF SCAFFOLDING

To assess multi-motif scaffolding in Genie 2, we curated 6 tasks where each requires multiple motifs, ranging from designing an immunogen with two epitopes (Yang et al., 2021) to scaffolding two $Ca^{2+}$ binding sites (four EF hand motifs) (Wang et al., 2022; Fallon and Quiocho, 2003). This set is meant to reflect the breadth of potential design problems, including immunogen, binder, and enyzme design. More details are included in Appendix C.

Genie 2 solves 4 of the 6 tasks. Figure 5A shows designs for task 1PRW_four (scaffolding with four $Ca^{2+}$ binding sites) and 4JHW+5WN9 (scaffolding with RSV-F site II and RSV-G 2D10 epitopes). More results are in Appendix G. We also apply Genie 2 to a multi-motif task proposed by Castro et al. (2024), which scaffolds an immunogen containing three unique epitopes from the respiratory syncytial virus (RSV) fusion protein. Genie 2 solves the task with only 1,000 samples (Figure 5B).

## 6 LIMITATIONS AND FUTURE WORK

Genie 2 achieves state-of-the-art performance on both unconditional generation and motif scaffolding. Yet, it takes longer to sample than other methods, requiring 1,000 denoising iterations vs. 100 (FrameFlow), 500 (Chroma), and 50 (RFDiffusion). Appendix H summarizes sampling times across sequence lengths. One future direction is thus to improve sampling efficiency. Genie 2 also employs triangular multiplicative updates, which scale cubically with sequence length, disproportionately affecting larger design tasks. A second future direction is thus to reduce the time and space complexity of Genie 2's architecture to enable generation of and training on larger proteins.

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

# A    ADDITIONAL DETAILS ON GENIE 2

## A.1    FRENET-SERRET FRAME CONSTRUCTION

Let $\mathbf{x}^i$ denotes the $C_\alpha$ coordinate at residue $i$. Following Lin and AlQuraishi (2023), we construct the Frenet-Serret frame at residue $i$ (denoted as $\mathbf{F}^i$) as

$$\mathbf{t}^i = \frac{\mathbf{x}^{i+1} - \mathbf{x}^i}{\|\mathbf{x}^{i+1} - \mathbf{x}^i\|} \qquad \mathbf{n}^i = \mathbf{b}^i \times \mathbf{t}^i$$
$$\mathbf{R}^i = \left[\mathbf{t}^i, \mathbf{b}^i, \mathbf{n}^i\right]$$
$$\mathbf{b}^i = \frac{\mathbf{t}^{i-1} \times \mathbf{t}^i}{\|\mathbf{t}^{i-1} \times \mathbf{t}^i\|} \qquad \mathbf{F}^i = (\mathbf{R}^i, \mathbf{x}^i)$$

For the frames of terminal residues, the first residue is assigned the same frame as the second residue, while the last residue is assigned the same frame as the second-to-last residue.

## A.2    HYPERPARAMETER CHOICES

In Table 2 we detail the key hyperparameters of the Genie 2 architecture and highlight differences from the original Genie model. For Genie 2, we increase input embedding and single representation dimensions as we found this improves performance without substantially impacting training speed. Due to the increase in model complexity, Genie 2 consists of 15.7M trainable parameters, $\sim$4x the original Genie architecture. However, Genie 2 remains four times smaller than RFDiffusion, which has 59.8M trainable parameters.

Table 2: Key hyperparamters of Genie and Genie 2. Updated values are indicated in **bold**.

| Hyperparameter | | Genie | Genie 2 |
| --- | --- | --- | --- |
| Number of parameters | | 4.1M | **15.7M** |
| Input embedding dimension | Residue index | 128 | **256** |
| | Chain index | - | **64** |
| | Diffusion timestep | 128 | **512** |
| Representation dimension | Single representation | 128 | **384** |
| | Pair representation | 128 | 128 |
| SE(3)-equivariant decoder | Number of IPA layers | 5 | **8** |

## A.3    TRAINING

For training, we use the Adam (Kingma and Ba, 2014) optimizer with a constant learning rate of $10^{-4}$. We train Genie 2 using data parallelism on 8 Nvidia A100 GPUs with an effective batch size of 48. We train the model for 40 epochs ($\sim$5 days) for a total of $\sim$960 GPU hours. In comparison, RFDiffusion is initialized with pretrained weights from RoseTTAFold (Baek et al., 2021), whose training requires 64 Nvidia V100 GPUs for 4 weeks. Training of RFDiffusion takes 3 days on 8 Nvidia A100 GPUs. Hence, Genie 2 requires much less computational resources to train than RFDiffusion.

## A.4    SAMPLING

To improve designability, we adjusted the sampling noise scale ($\gamma$ in Equation (3)) to trade diversity for designability. We set $\gamma = 0.6$ and $\gamma = 0.4$ for unconditional generation and motif scaffolding, respectively, as these settings provided the best results. Moreover, for motif scaffolding, we use the checkpoint at epoch 30 since it gives slightly better performance. Appendix E.4 provides further information on the selection of model checkpoints and Appendix E.3 provides further discussions on the effect of sampling noise scale.

## A.5 ADDITIONAL BENCHMARKING DETAILS

For competing methods, we use their pretrained weights, together with their default hyperparameter settings, provided in their GitHub repositories. Specifically for sampling noise scale, RFDiffusion samples with 50 steps at a noise scale of 1; Proteus and CarbonNovo sample with 100 steps at a noise scale of 0.1. Chroma, FrameFlow and MultiFlow follow different sampling procedures and thus sampling noise scale is inapplicable.

# B  ABLATION STUDIES

In this section we provide ablation studies on Genie training. To evaluate models, we follow the same procedure from Section 4.2 when assessing unconditional generation performance: for each model, we generate 5 samples per sequence length ranging from 50 to 256 residues. For single-motif scaffolding evaluation, we use the pipeline described in Section 5.2, but sample only 100 designs per motif scaffolding problem to minimize computational costs.

## B.1  EFFECT OF VARYING CONDITIONAL TASK RATIO

We experimented with varying the frequency of conditional vs. unconditional tasks during training (0.0, 0.2, 0.5, 0.8, and 1.0). Due to computational constraints, we trained models only up to 10 epochs. Although models do not fully converge, we believe the trends are still indicative of final performance.

Table 3 summarizes the performance of these models on both unconditional protein generation and single motif scaffolding. As the conditional task ratio increases, motif scaffolding performance generally improves, with the best performance achieved when the conditional task ratio equals 1. Surprisingly, unconditional generation performance fluctuates but is ultimately also maximized when conditional tasks are exclusively sampled. As a result we use a conditional task ratio of 1 during all training runs.

Table 3: Unconditional generation and motif scaffolding performance by conditional task ratio. SUCCESSES denote total number of unique successes across all problems.

| RATIO | UNCONDITIONAL GENERATION | | | MOTIF SCAFFOLDING | |
|---|---|---|---|---|---|
| | DESIGNABILITY | DIVERSITY | $F_1$ | SOLVED | SUCCESSES |
| 0.0 | 0.858 | 0.771 | 0.812 | 1 | 6 |
| 0.2 | 0.892 | 0.752 | 0.816 | 14 | 101 |
| 0.5 | 0.740 | 0.649 | 0.692 | 13 | 98 |
| 0.8 | 0.865 | 0.783 | 0.822 | 18 | 179 |
| **1.0** | **0.898** | **0.802** | **0.847** | **19** | **202** |

## B.2 EFFECT OF NUMBER OF DIFFUSION STEPS

We experimented with using a different number of diffusion steps during training (100, 200 and 500). We trained each model for 40 epochs and compared their performance with the model in the main text, which is trained with 1,000 diffusion steps. Table 4 summarizes performance on both unconditional generation and motif scaffolding. For consistency, we use a sampling noise scale of 0.6 for unconditional generation and 0.4 for motif scaffolding (same as the main text).

While Genie 2 achieves the best performance on both unconditional generation and motif scaffolding when trained with 1,000 diffusion steps, its performance is comparable when trained with fewer diffusion steps (number of diffusion steps during generation is always matched to that used during training). When compared to existing state-of-the-art protein diffusion models (Table 1), Genie 2 achieves comparable designability but higher diversity even when trained with as few as 100 diffusion steps. Thus, Genie 2 provides support for fast inference with minimal loss of the model's generative capability. Detailed sampling time statistics for Genie 2 trained with fewer diffusion time steps are included in Table 14; compared to other models, Genie 2 trained with 100 diffusion step has comparable sampling efficiency when generating long proteins, but much higher sampling efficiency when generating short proteins.

Table 4: Unconditional generation and motif scaffolding performance by number of diffusion steps. SUCCESSES denote total number of unique successes across all problems. In the last row, we include the performance of RFDiffusion for reference.

| DIFFUSION STEPS | UNCONDITIONAL GENERATION | | | MOTIF SCAFFOLDING | |
|---|---|---|---|---|---|
| | DESIGNABILITY | DIVERSITY | $F_1$ | SOLVED | SUCCESSES |
| 100 | 0.90 | 0.82 | 0.86 | 20 | 332 |
| 200 | 0.95 | 0.70 | 0.81 | **21** | 252 |
| 500 | 0.89 | 0.81 | 0.85 | 19 | 291 |
| 1000 | **0.96** | **0.91** | **0.93** | **21** | **345** |
| RFDIFFUSION | **0.96** | 0.63 | 0.76 | **21** | 223 |

### B.3 COMPARISON WITH PDB-TRAINED GENIE 2

We train and assess a version of Genie 2 trained exclusively on the PDB, unlike the model described in the main text which is trained on the clustered AFDB database. To train Genie 2 on the PDB, we obtain monomeric PDB structures with a cutoff date of April 2, 2024. We further filter these structures to have a maximum sequence length of 256 and a minimum resolution of 5Å while discarding structures with missing $C_\alpha$ atom coordinates. This results in 17,970 structures. To simplify comparison, we use the same set of hyperparameters described in Appendix A.2 and train the model on the PDB dataset for 1,600 epochs, which is equivalent in number of training iterations to ~40 epochs when training on the clustered AFDB dataset. For assessment, we use the 800th epoch checkpoint since it gives slightly better performance and we sample at a noise scale of 0.6 (same as the main text). Table 5 compares the performance of Genie 2 trained on the PDB with the performance of Genie 2 trained on AFDB, demonstrating better generation performance by the latter model.

Table 5: Unconditional generation and motif scaffolding performance by training dataset. EPOCHS denote the number of training epochs. SUCCESSES denote total number of unique successes across all problems.

| DATASET | EPOCHS | UNCONDITIONAL GENERATION | | | MOTIF SCAFFOLDING | |
|---|---|---|---|---|---|---|
| | | DESIGNABILITY | DIVERSITY | $F_1$ | SOLVED | SUCCESSES |
| PDB | 800 | 0.736 | 0.630 | 0.679 | 19 | 194 |
| AFDB | 20 | 0.892 | 0.804 | 0.846 | 19 | 270 |
| AFDB | 40 | **0.958** | **0.905** | **0.931** | **21** | **345** |

## C  MULTI-MOTIF SCAFFOLDING BENCHMARK

In Table 6, we provide detailed configurations for each multi-motif scaffolding task. We name each problem using the names of PDB structures that contain the motif(s) used in the problem. Additional postfixes are added to distinguish between problems whose motifs come from the same PDB structure. In the third column ("configuration"), we provide a detailed input specification for each multi-motif scaffolding problem. Each bolded part denotes a motif segment, including its location in the PDB structure. For example, "5WN9/A170-189{2}" in problem 4JHW+5WN9 indicates that the motif segment comes from residue 170 - 189 of Chain A in the protein 5WN9, and "2" (in curly bracket) indicates that this motif segment belongs to the second motif. Each non-bolded part denotes a scaffold segment with minimum and maximum lengths specified. At sampling time, each scaffold length is sampled within this range. For example, "10-40" in problem 4JHW+5WN9 indicates that the scaffold has a length between 10 and 40 (inclusive). The last column ("Total length") specifies the minimum and maximum length requirements for the whole sequence.

Table 6: The benchmark set of multi-motif scaffolding problems.

| Name | Description | Configuration | Length |
|---|---|---|---|
| **4JHW+5WN9** (Yang et al., 2021) | Two epitopes | 10-40, **4JHW/F254-278{1}**, 20-50, **5WN9/A170-189{2}**, 10-40 | 85-175 |
| **1PRW_two** (Wang et al., 2022; Fallon and Quiocho, 2003) | Two 4-helix bundles | 5-20, **1PRW/A16-35{1}**, 10-25, **1PRW/A52-71{1}**, 10-30, **1PRW/A89-108{2}**, 10-25, **1PRW/A125-144{2}**, 5-20 | 120-200 |
| **1PRW_four** (Wang et al., 2022; Fallon and Quiocho, 2003) | Four EF-hands | 5-20, **1PRW/A21-32{1}**, 10-25, **1PRW/A57-68{2}**, 10-25, **1PRW/A94-105{3}**, 10-25, **1PRW/A125-144{4}**, 5-20 | 88-163 |
| **3BIK+3BP5** (Bryan et al., 2021) | Two PD-1 binding motifs | 5-15, **3BIK/A121-125{1}**, 10-20, **3BP5/B110-114{2}**, 5-15 | 30-60 |
| **3NTN** (Agnew et al., 2011; Chalkley et al., 2022) | Two 3-helix bundles | 5-15, **3NTN/A342-348{1}**, 10-10, **3NTN/A367-372{2}**, 10-20, **3NTN/B372-367{2}**, 10-10, **3NTN/B348-342{1}**, 10-20, **3NTN/C342-348{1}**, 10-10, **3NTN/C367-372{2}**, 5-15 | 99-139 |
| **2B5I** (Ren et al., 2022; Silva et al., 2019) | Two binding sites | 5-15, **2B5I/A11-23{2}**, 10-20, **2B5I/A35-45{1}**, 10-20, **2B5I/A61-72{1}**, 5-15, **2B5I/A81-95{2}**, 20-30, **2B5I/A119-133{2}** | 116-166 |

For problem 3NTN, the original PDB structure is a homotrimer. It consists of three helices, which together form a binding site for $Ni^{2+}$ ion and a binding site for $Cl^-$ ion. When setting up this multi-motif scaffolding problem, we are interested in whether it is possible to combine two binding sites (formed by multiple chains) into a single-chain protein. One possible reason that Genie 2 fails on this task might be that this problem is not solvable given the current specification.

# D DISCUSSION ON SELF-CONSISTENCY PIPELINE

## D.1 LIMITATIONS OF SELF-CONSISTENCY PIPELINE

While the self-consistency pipeline we employ is a widely used *in silico* evaluation pipeline for protein design, it cannot replace experimental validations due to the complexity involved in functional protein design (for example, conformational changes), most of which are not captured by existing computational models. However, given the expense and difficulty of experimental validation, self-consistency pipelines can serve as an initial filter for potential design candidates to improve experimental success rates.

In addition, while self-consistency pipelines can be reliable at assessing the designability of rigid structures (more specifically, structures with more than 50% secondary structure contents), they are less reliable at assessing flexible structures due to the lack of a single correct solution and the inability of current structure prediction models to predict conformational ensembles. This potential flexibility of designed proteins could thus deflate self-consistency metrics, resulting in inaccurate assessment of designability. Some existing methods, such as RFDiffusion, have opted to train only on PDB structures with at least 50% secondary structure content. In Genie 2 we do not take this approach, training on the entirety of AFDB including potentially flexible structures, and this may why Genie 2 exhibits low designability at the normal sampling noise scale (more details in Appendix D.3). By learning to model the data distribution of AFDB structures, Genie 2 generates more flexible structures at normal sampling noise scale (Figure 2B), but whether these structures are viable or not cannot be assessed by existing self-consistency pipelines.

## D.2 EFFECT OF DIFFERENT INVERSE FOLDING MODELS

Figure 6 visualizes the scRMSD distribution across sequence lengths using different inverse folding models. We observe that our self-consistency pipeline behaves similarly regardless of whether ProteinMPNN or ESM-IF is used. This suggests that even though ProteinMPNN is trained only on PDB structures (compared to ESM-IF, which is also trained on AFDB structures), it does not exhibit a bias towards PDB structures. We also observe that our self-consistency pipeline performs differently between PDB and AFDB structures; a detailed analysis on this observation is provided in Appendix D.3.

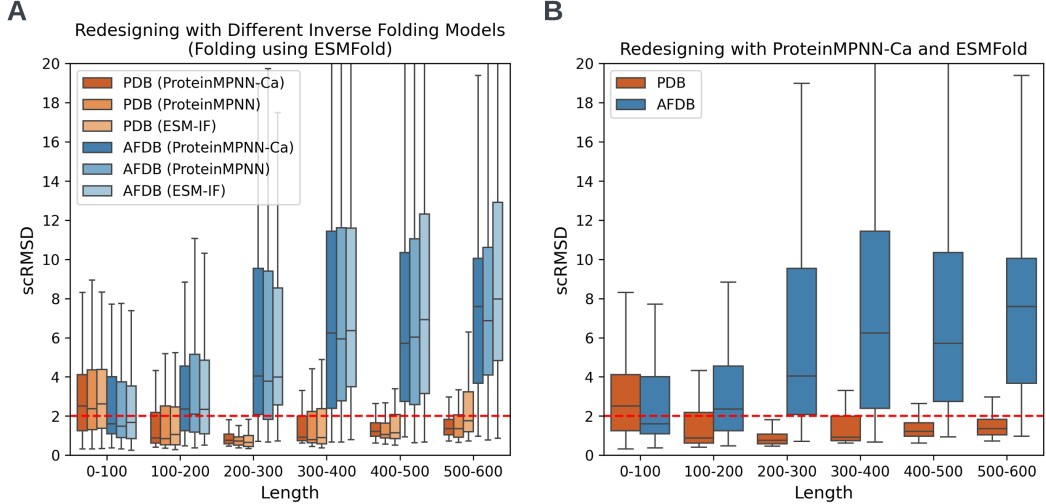

Figure 6: Behavior of the self-consistency pipeline when using different inverse folding methods and randomly sampled structures from the PDB versus AFDB. For each dataset/length combination, 100 structures are sampled. (A) Distribution of scRMSD across sequence lengths using different inverse folding models. (B) Distribution of scRMSD across sequence lengths using ProteinMPNN-Ca and ESMFold as the self-consistency evaluation pipeline.

### D.3 Differences in self-consistency evaluations for PDB and AFDB structures

In Table 2C, we observe a discrepancy in designability between PDB and AFDB structures. To better understand this phenomenon, we visualize the distribution of scRMSD as a function of sequence length in Figure 6B, where 100 structures are sampled for each length bucket. The scRMSD distributions for AFDB structures are generally above that for PDB structures across all lengths, and the difference increases with sequence length.

One hypothesis for this observation is that proteins tend to contain multiple domains as their lengths increase (most single domains range in length between 40 and 200 residues). For a multi-domain AFDB structure, while each domain may be compact, the linkers that connect its domains can be flexible, leading to flexibility in the structure of the whole protein and to higher protein radii. PDB structures, in contrast, are likely enriched for proteins with more stable inter-domain orientations, either due to crystallization conditions which fix multi-domain proteins into a single conformation or due to the proteins themselves being inherently less flexible as such proteins are easier to crystallize. Regardless of the root cause, these factors will result in an experimental bias towards less flexible structures and smaller protein radii (compared to AFDB structures). We see confirmation of this when visualizing the distribution of protein radii across lengths in Figure 7; for each length bucket, 100 structures are randomly sampled for evaluation. We observe that as sequence length increases, differences in protein radii increase between AFDB and PDB structures, consistent with our hypothesis.

To further substantiate this hypothesis, we visualize an example of an AFDB multi-domain structure (AFDB ID: Q6GIK1, Figure 8A) and its redesign using ESMFold in Figure 8: while individuals domain are redesigned accurately using ESMFold (Figure 8C), the flexible linker results in mis-alignment between the original and redesigned structures, leading to high scRMSD and thus low designability.

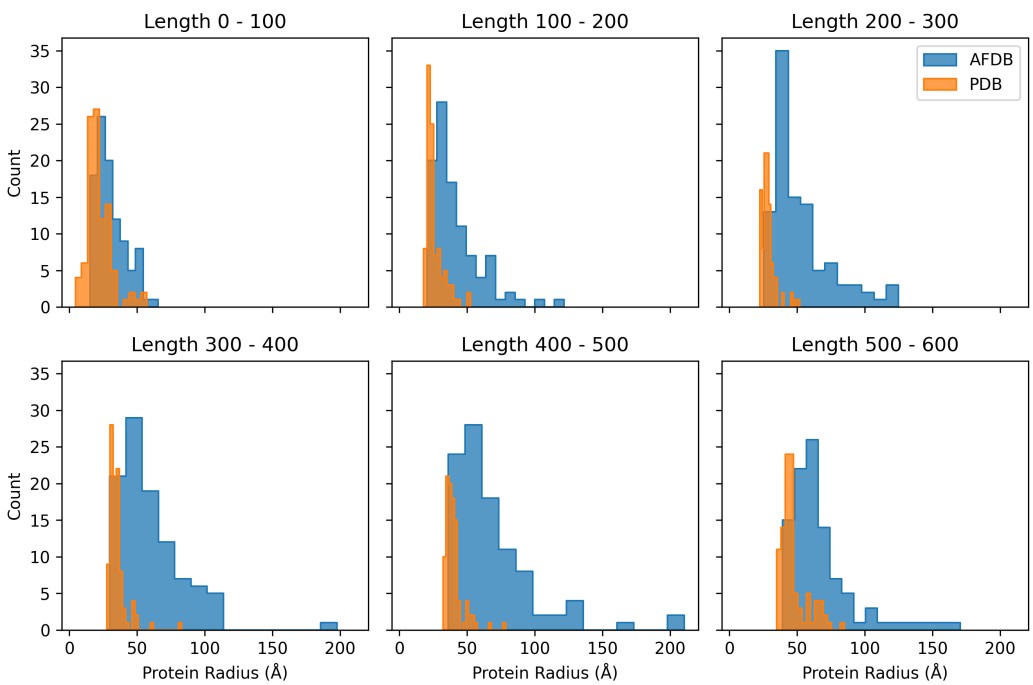

Figure 7: Distribution of protein radii across different sequence lengths. For each length bucket, 100 structures are randomly sampled from each dataset.

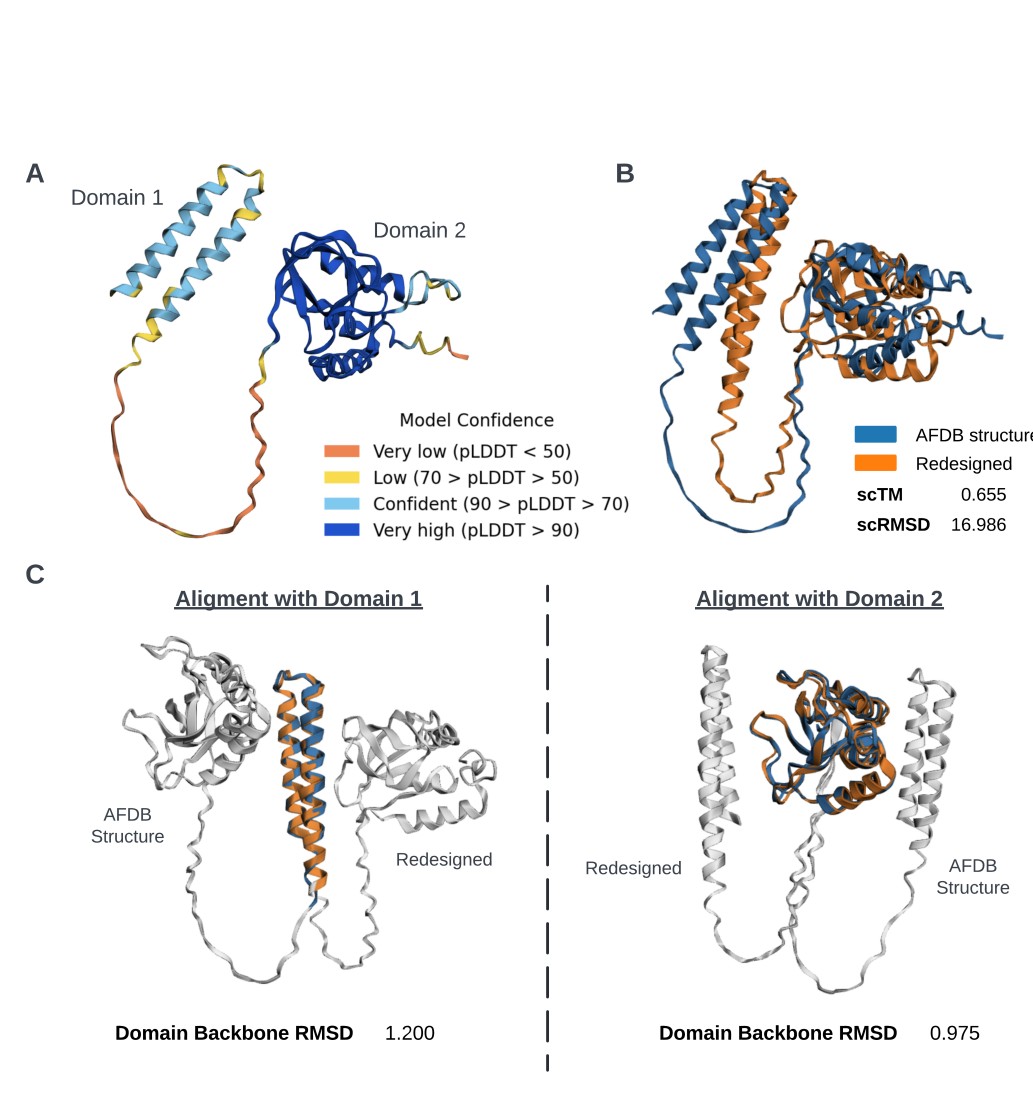

Figure 8: Visualization of a multi-domain AFDB structure (AFDB ID: Q6GIK1) and its ESMFold-redesigned structure. **(A)** Visualization of AFDB structure with pLDDT overlaid. **(B)** AFDB structure with its ESMFold-redesigned structure aligned using Kabsch's algorithm. **(C)** Alignments of AFDB structure and its ESMFold-redesigned structure based on domain 1 (left) and domain 2 (right).

# E  ADDITIONAL RESULTS ON UNCONDITIONAL PROTEIN GENERATION

## E.1  COMPARISON WITH GENIE

In this section we compare against the original version of Genie (Lin and AlQuraishi, 2023), specifically Genie-SwissProt. We include Genie-SwissProt's performance on unconditional generation in Table 7 for reference. We use Genie-SwissProt since it is trained on AlphaFold-SwissProt database for 100 epochs and achieves the best performance in both designability and diversity out of all the original Genie variants we considered. As the evaluation metrics for Genie-SwissProt are consistent with those in this manuscript, we directly use the numbers that are reported in the Genie paper. We note that, strictly speaking, it is unfair to compare Genie to Genie 2 due to differences in training dataset, model architecture, and model complexity. Nonetheless, we find the comparison informative, with the caveat that there are multiple differences between the two models that extend beyond architecture.

Table 7: Unconditional generative performance of Genie and Genie 2.

| METHOD | DESIGNABILITY | DIVERSITY | $F_1$ | PDB NOVELTY |
|---|---|---|---|---|
| GENIE | 0.79 | 0.64 | 0.71 | 0.04 |
| GENIE 2 | **0.96** | **0.91** | **0.93** | **0.41** |

## E.2  SIMILARITY TO PDB AND AFDB STRUCTURES

Figure 9 visualizes the distribution of TM scores to the most similar structure in the reference dataset (PDB/AFDB) and Figure 9B visualizes the scatterplot of TM scores with respect to the most similar PDB and AFDB structures (the same set curated for analysis in Table 1). We observe that structures generated by Genie 2 are more similar to AFDB structures than PDB structures; this is as expected since Genie 2 is trained to learn the data distribution of AFDB structures.

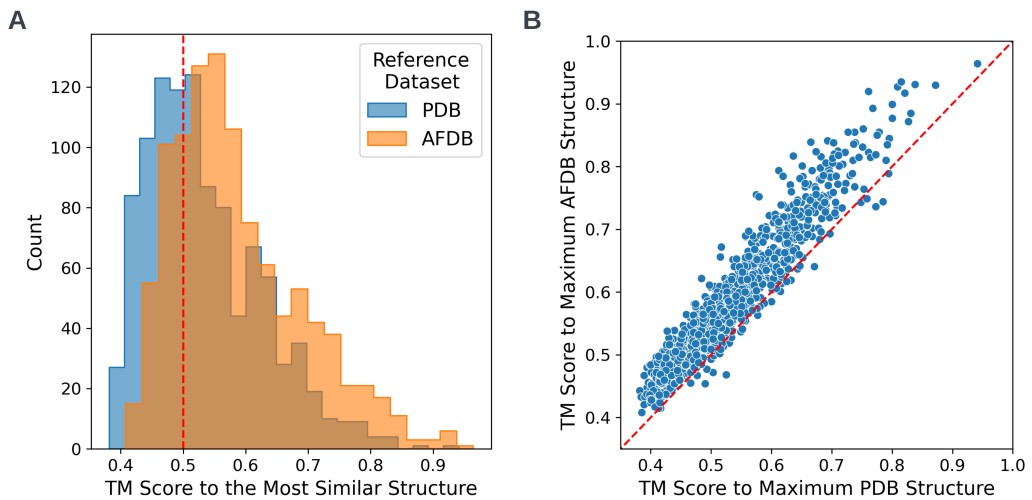

Figure 9: Similarity to PDB vs. AFDB structures. **(A)** Distribution of TM scores to the most similar structure in PDB and AFDB. **(B)** Scatterplot of TM scores with respect to the most similar PDB/AFDB structures.

### E.3 EFFECT OF SAMPLING NOISE SCALE

Lowering sampling noise scale has a similar effect to sampling at lower temperature, which trades diversity for higher designability. However, there is one minor caveat: the noise schedule, or noise variance at each diffusion step, is predefined as a function of diffusion time step. When lowering the sampling noise scale, we reduce the variance of noise in the reverse process, which deviates from the noise schedule that is predefined by the diffusion time steps. Since the diffusion time step is used as an input condition for the denoiser, such deviation in noise variance implicitly causes the denoiser to consider each input structure to be noisier than it actually is. This could potentially help the model to correct for its accumulated errors from prior reverse steps and thus generate structures with higher quality. However, the mismatch between the noise variance of the input structure and the model-perceived noise variance (based on the diffusion time step) also leads model inputs to be out of training distribution. This may deteriorate generative performance. Hence, we consider sampling noise scale as a hyperparameter to be tuned and profile model performance on unconditional generation across different noise scales. The results are summarized in Table 8, and indicate that a noise scale of 0.6 allows the model to generate the most diverse set of structures. Here, for each sampling noise scale, we generate 5 structures per length between 50 and 256 (1,035 structures in total).

Table 8: Unconditional generation performance by sampling noise scale.

| NOISE SCALE ($\gamma$) | DESIGNABILITY | DIVERSITY | $F_1$ |
|---|---|---|---|
| 0.0 | 0.636 | 0.192 | 0.295 |
| 0.2 | 0.840 | 0.545 | 0.661 |
| 0.4 | **0.977** | 0.856 | 0.912 |
| 0.6 | 0.958 | **0.905** | **0.931** |
| 0.8 | 0.693 | 0.621 | 0.655 |
| 1.0 | 0.141 | 0.129 | 0.134 |

An alternative way to interpret the effect of sampling noise scale is to analyze the radius of generated structures across different sampling noise scales. Here, we consider the radius of a protein (represented by a trace of $C_\alpha$ atoms) as the distance between the center and the furthest $C_\alpha$ atom. Figure 10A visualizes the distribution of protein radii across different sampling noise scales (same set of generated structure as Table 8) and Figure 10B visualizes the distribution of protein radii for 1,000 structures randomly sampled from our filtered AFDB training set (with a maximum sequence length of 256 residues). We observe that when sampling at the normal noise scale ($\gamma = 1$), the distribution of protein radii resembles that of the AFDB training set. While this implies that Genie 2 learns to model the training data distribution, it also suggests that Genie 2 learns to generate non-compact proteins similar to those in AFDB, which has low designability based on our *in silico* self-consistency pipeline. This further explains why Genie 2 performs poorly when the sampling noise scale is set to 1. A lower sampling noise scale contracts the expansion of protein radius in the reverse diffusion process (as shown in Figure 10A), encouraging the generation of compact protein structures that are more likely to be designable.

We additionally visualize the secondary structure distribution resulting from different sampling noise scales in Figure 11. We observe that as the sampling noise scale decreases, secondary structure diversity also decreases; this drop in diversity is likely due to the contraction of protein radii which encourages more compact protein structures. Such contraction thus indirectly promotes a higher content of secondary structures, or more specifically helices, which are more condensed than extended strands. Hence, we observe that as sampling noise scale decreases, the secondary structure distribution is shifted towards regions with higher helical content.

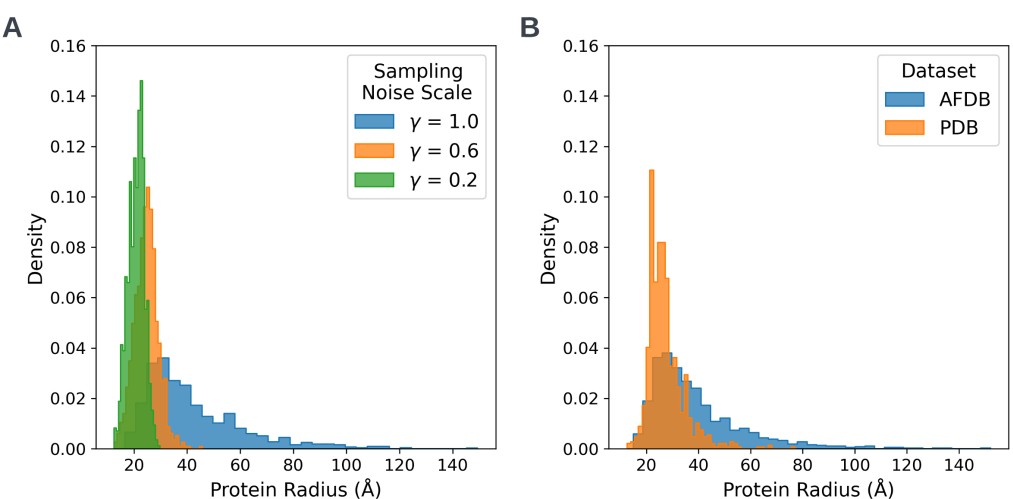

Figure 10: Distribution of protein radii. (A) Distribution of protein radii for Genie 2-generated structures as a function of sampling noise scale. (B) Distribution of protein radii for 1,000 randomly sampled structures from PDB and AFDB (with a maximum sequence length of 256 residues to be consistent with the training setup).

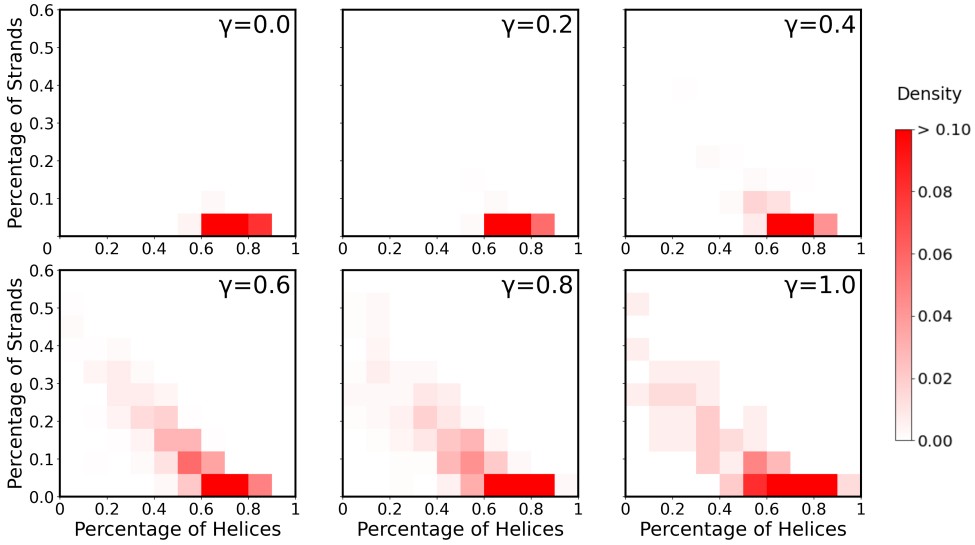

Figure 11: Secondary structure distribution of structures by sampling noise scale.

### E.4 Genie 2 Can Design Longer Proteins When Trained for Longer

Here we investigate how Genie 2's performance varies as a function of the number of training epochs. Figure 12 visualizes Genie 2's unconditional generation performance as a function of protein length and across different model checkpoints. For each length, 100 samples are generated and evaluated using the self-consistency pipeline used in Figure 3. As training continues, Genie 2 significantly improves on generating diverse longer proteins, surpassing existing methods such as Proteus and RFDiffusion (in terms of diversity). It is worth noting that as training continues, the performance on short protein generation slightly declines. This suggests that during training, the model starts by learning to generate short proteins (an easier task) before tackling the generation of longer proteins (more challenging task). The decline in diversity of short protein generation could be the result of the model allocating more resources to modeling longer proteins.

For our unconditional generation analysis in the main text, we select the 40 epoch checkpoint since it achieves better performance for our in-distribution analysis (on proteins with 50 - 256 residues). However, Genie 2 is capable of generating more diverse proteins with more than 256 residues than reported in the main text (by using the 50 epoch checkpoint) and the best generative performance of Genie 2 could be achieved through a length-based ensemble of Genie 2 training checkpoints (for example, by using the 40-epoch checkpoint for proteins with at most 256 residues and the 80-epoch checkpoint for proteins with more than 256 residues). Such ensembling would not affect sampling efficiency since the checkpoint is selected before sampling based on desired protein length. We provide a wrapper for this functionality in the Genie 2 repository.

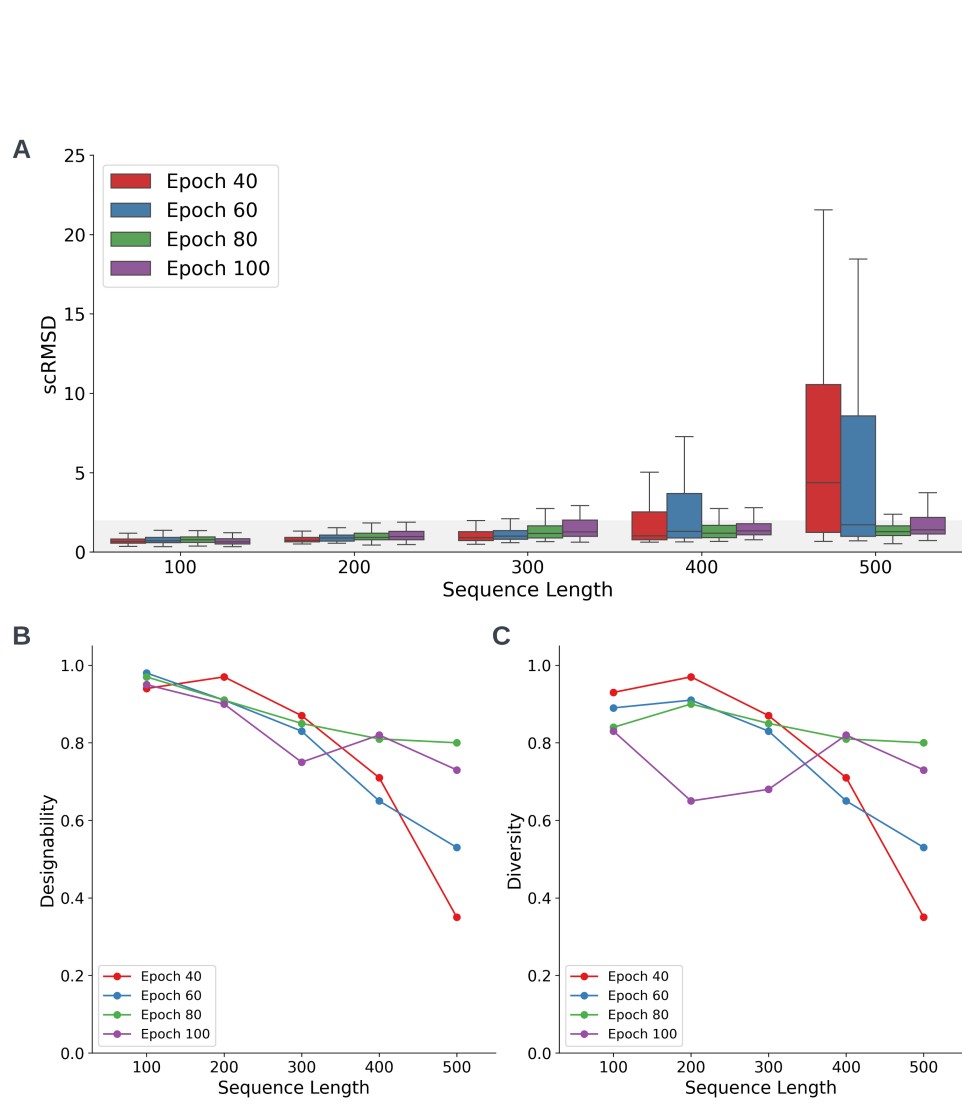

Figure 12: Length-based assessment of Genie 2 across different model checkpoints. For each checkpoint/sequence length combination, we generate 100 structures. **(A)** Box-and-whisker plots of scRMSDs between generated structures and their most similar ESMFold-predicted structures. **(B-C)** Plots of designability (B) and diversity (C) as a function of sequence length, with the same color scheme as (A).

### E.5 LENGTH-BASED PERFORMANCE ANALYSIS USING SCTM

We provide additional assessments of Genie 2 and competing methods using a second designability metric, the self-consistency TM score (scTM). scTM is computed using the same pipeline as scRMSD, described in Section 4.1, except using TM score to measure the structural distance between a generated structure and its most similar ESMFold-predicted structure. scTM is a less stringent metric than scRMSD since TM score is less sensitive to minor structural variations. Figure 13A visualizes the distribution of scTM by sequence length for Genie 2 and competing methods, while Figures 13B and 13C visualize scTM-based designability and diversity as a function of sequence length, respectively. Here, a structure is considered as scTM-based designable if it satisfies both scTM $> 0.5$ and pLDDT $> 70$. Diversity is computed using the same clustering procedure described in section 4.1. Overall trends remain consistent with our main results.

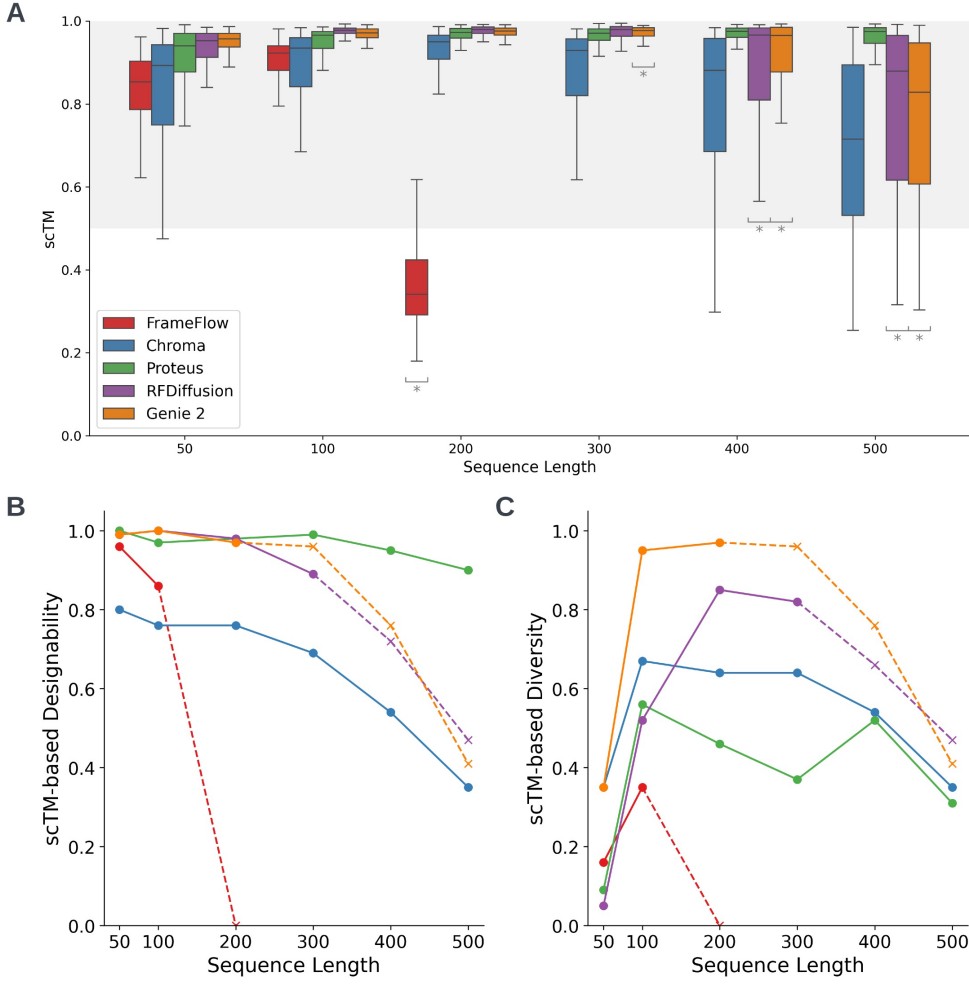

Figure 13: Assessment of Genie 2 and competing methods by sequence length using scTM as the designability metric. 100 structures are generated per sequence length and method. **(A)** Distribution of self-consistency TM between generated structures and the most similar ESMFold-predicted structures. Asterisk (*) indicates that the sampled sequence length is beyond the maximum sequence length sampled at training time. **(B)** Plot of scTM-based designability (percentage of scTM-designable structures) as a function of sequence length, with the same color scheme as (A). Out-of-distribution generation is indicated with crosses and dashed lines. **(C)** Plot of scTM-based diversity (percentage of unique scTM-based designable clusters) as a function of sequence length, with the same color scheme as (A). Out-of-distribution generation is indicated with crosses and dashed lines.

## E.6 ANALYSIS OF DESIGNABILITY AS A FUNCTION OF HELICITY

We use the set of Genie 2-generated structures sampled for analysis in Table 1 and visualize their scRMSD distribution as a function of fraction of helices (Figure 14A) and strands (Figure 14B). While scRMSD slightly decreases with higher helicity, the majority of generated structures have relatively low scRMSD independent of helicity. This, combined with the secondary structure distribution shown in Figure 2A, suggests that while Genie 2 has a tendency to generate more helical structures, its generation quality of helical and stranded structures is similar.

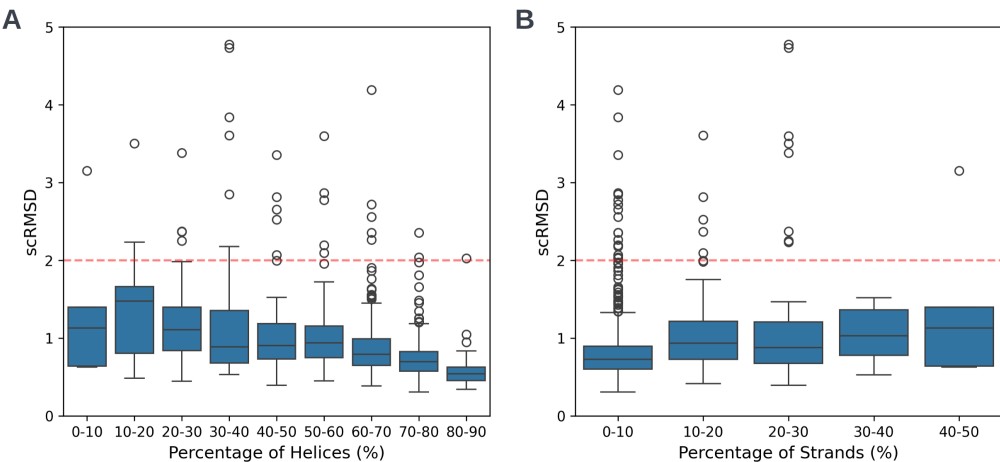

Figure 14: Distribution of scRMSD as a function of fraction of alpha helices (A) and beta strands (B). The dataset used for this assessment is the same dataset generated by Genie 2, which is used for assessment of unconditional generation performance in Table 1.

## E.7 COMPARISONS TO MULTIFLOW AND CARBONNOVO

We compared Genie 2 to MultiFlow (Campbell et al., 2024) and CarbonNovo (Ren et al.), both of which are capable of joint sequence-structure co-design. For MultiFlow, we use ProteinMPNN for sequence redesign since it gives better performance on both designability and diversity, while for CarbonNovo we use sequences predicted by the model. All sequences are folded using ESMFold and the consistency between ESMFold-predicted structures and designed structures is then computed. Similar to our assessment in Section 4.2, for each method, we generate 5 samples of every length ranging from 50 to 256 residues (1,035 structures in total). The results are summarized in Table 9, which show similar trends as in the main text. While MultiFlow and CarbonNovo achieve similar performance to Genie 2 on designability, Genie 2 substantially outperforms both on diversity and novelty.

Table 9: Comparison of unconditional generation between MultiFlow, CarbonNovo, and Genie 2.

| METHOD | DESIGNABILITY | DIVERSITY | $F_1$ | PDB NOVELTY |
|---|---|---|---|---|
| MULTIFLOW | **0.98** | 0.61 | 0.75 | 0.18 |
| CARBONNOVO | 0.94 | 0.62 | 0.75 | 0.16 |
| GENIE 2 | 0.96 | **0.91** | **0.93** | **0.41** |

## E.8    ADDITIONAL EXAMPLES OF DESIGNABLE CLUSTERS BY GENIE 2

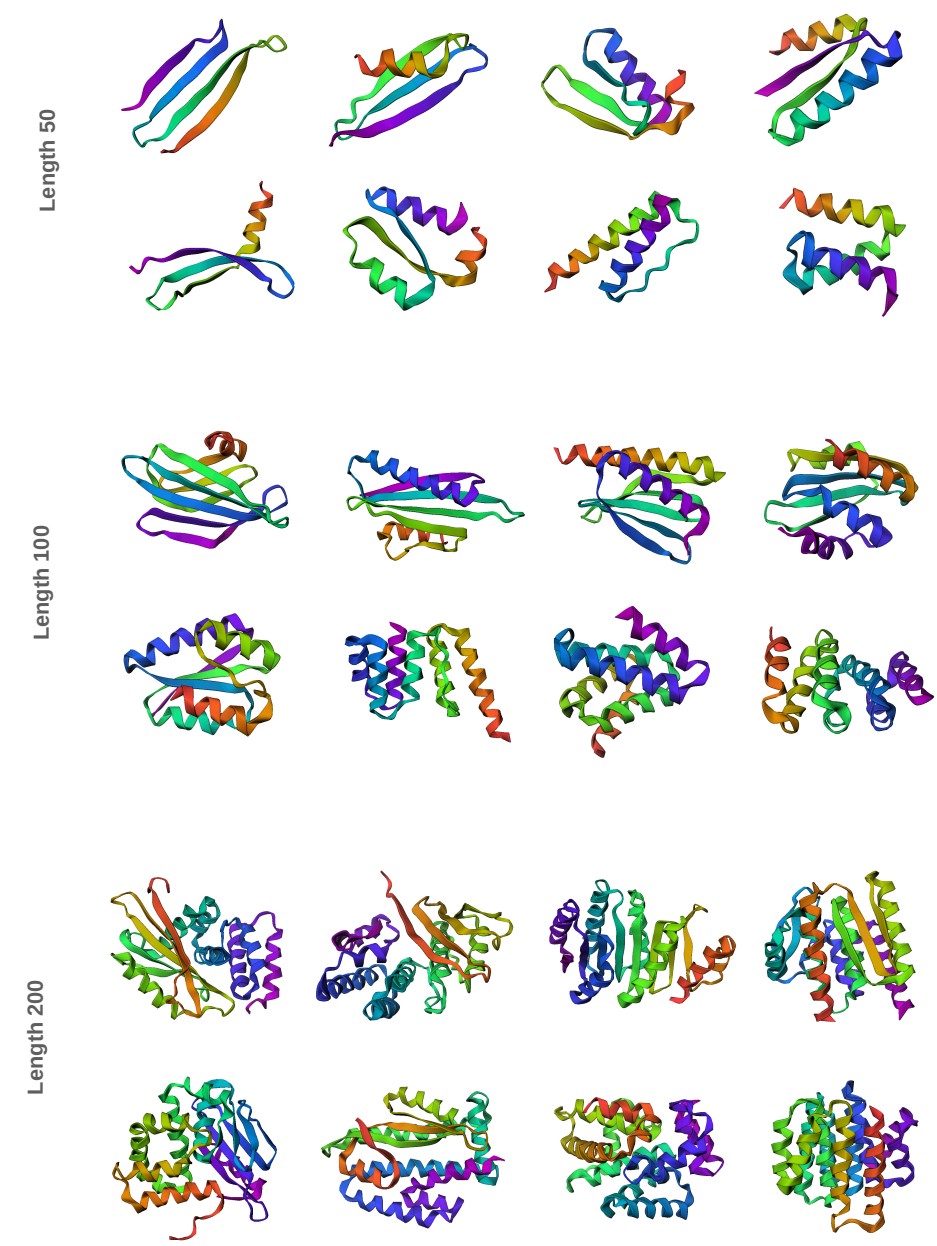

Figure 15: Examples of Genie 2 designed structures with in-distribution sequence lengths (within the maximum sequence length of 256 set at training time).

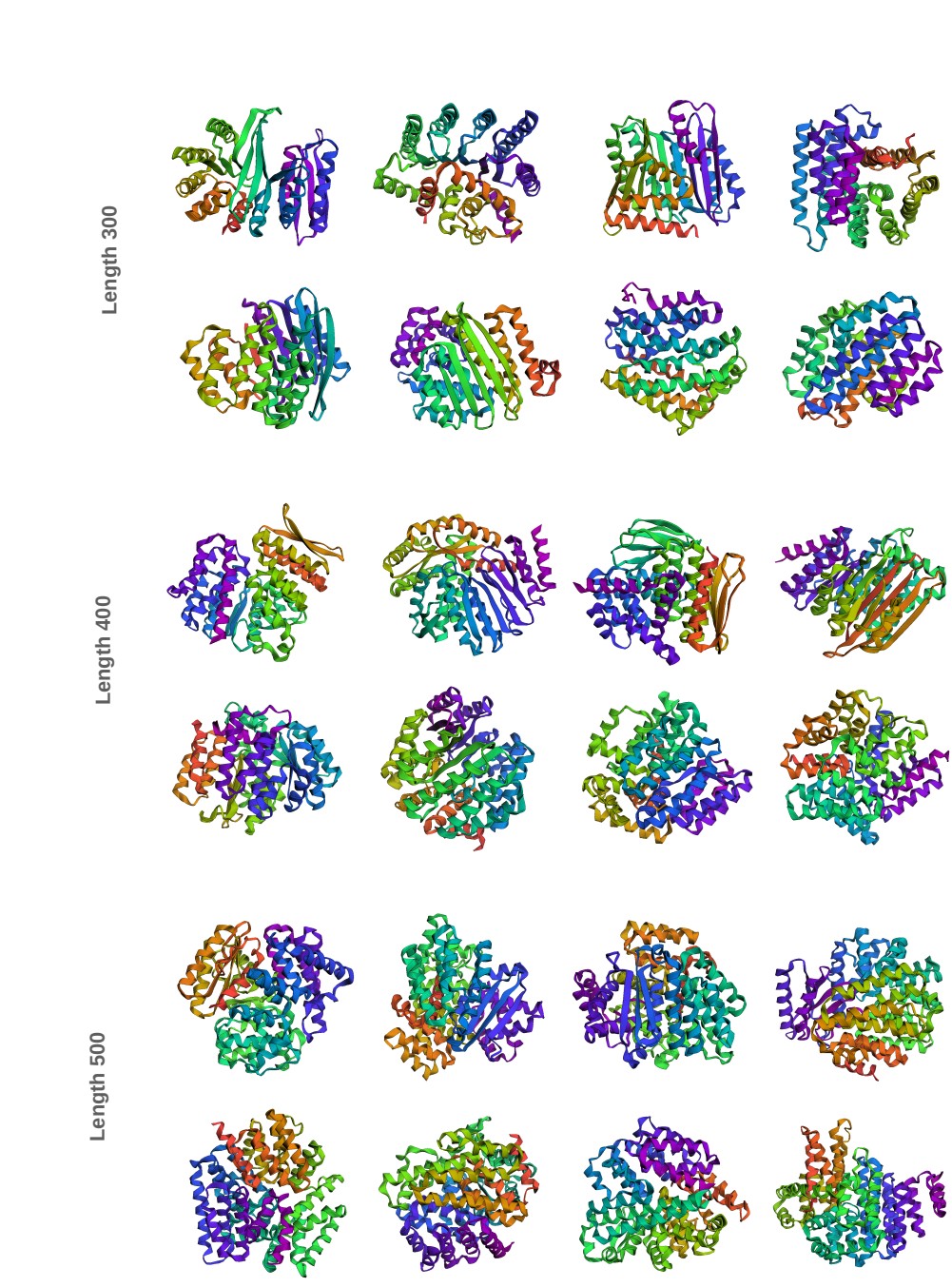

Figure 16: Examples of Genie 2 designed structures with out-of-distribution sequence lengths (longer than 256 residues).

## E.9 ADDITIONAL INFORMATION

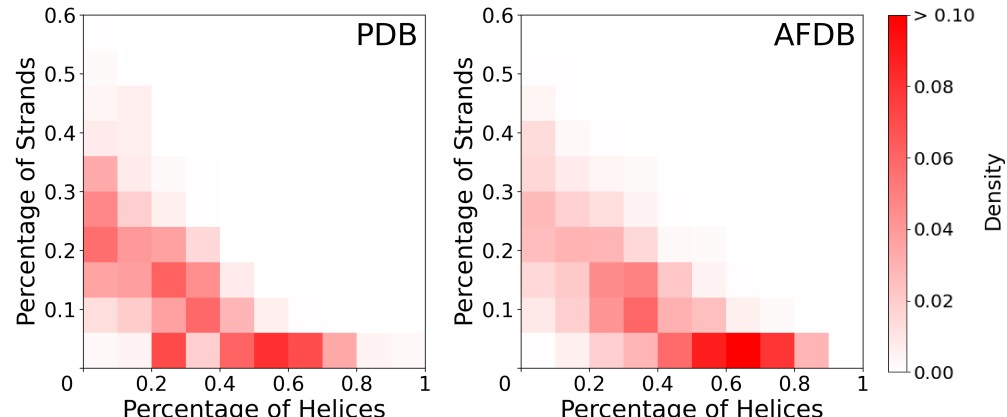

Figure 17: Secondary structure distribution of 1,000 randomly sampled structures from PDB (left) and AFDB (right).

# F    ADDITIONAL RESULTS ON SINGLE-MOTIF SCAFFOLDING

## F.1    EVALUATION DETAILS

Watson et al. (2023) asserts that RFDiffusion achieves a higher success rate when the noise scale is set to 0; however, this success rate does not account for the diversity of designed structures. To ensure a fair comparison, we first assessed the performance of RFDiffusion with noise scale set to 0 and 1. For each motif scaffolding problem, we sampled 100 structures per problem and evaluated them using the same pipeline as in Section 5.2. Figure 18 visualizes the number of unique successes by motif scaffolding problems. We observe that RFDiffusion solves more motif scaffolding problems with more diverse designs when the noise scale is set to 1. Thus, to maximize RFDiffusion's performance, we compare Genie 2 with RFDiffusion with a noise scale of 1 throughout this work.

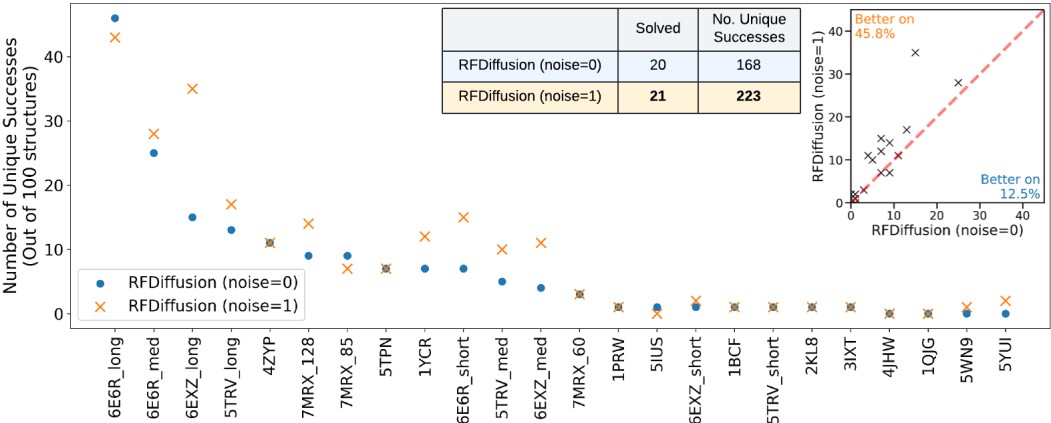

Figure 18: Performance of RFDiffusion with a noise scale of 0 and 1 across 24 single-motif scaffolding tasks. Inset (top right) shows a scatter plot of the (unique) success rate of RFDiffusion with a noise scale of 1 versus RFDiffusion with a noise scale of 0; each point represents a scaffolding task. Summary statistics are shown in table (left).

F.2 NUMBER OF UNIQUE SUCCESSES

Table 10: Number of unique successes (out of 1,000 structures) generated by Genie 2, RFDiffusion, and FrameFlow on each single-motif scaffolding task.

| Name | Genie 2 | RFDiffusion | FrameFlow |
|---|---|---|---|
| 6E6R_long | **415** | 381 | 110 |
| 6EXZ_long | 326 | 167 | **403** |
| 6E6R_med | **272** | 151 | 99 |
| 1YCR | 134 | 7 | **149** |
| 5TRV_long | **97** | 23 | 77 |
| 6EXZ_med | 54 | 25 | **110** |
| 7MRX_128 | 27 | **66** | 35 |
| 6E6R_short | **26** | 23 | 25 |
| 5TRV_med | **23** | 10 | 21 |
| 7MRX_85 | **23** | 13 | 22 |
| 3IXT | **14** | 3 | 8 |
| 5TPN | **8** | 5 | 6 |
| 7MRX_60 | **5** | 1 | 1 |
| 1QJG | 5 | 1 | **18** |
| 5TRV_short | **3** | 1 | 1 |
| 5YUI | **3** | 1 | 1 |
| 4ZYP | 3 | **6** | 4 |
| 6EXZ_short | 2 | 1 | **3** |
| 1PRW | **1** | **1** | **1** |
| 5IUS | **1** | **1** | 0 |
| 1BCF | **1** | **1** | 1 |
| 5WN9 | 1 | 0 | **3** |
| 2KL8 | **1** | **1** | 1 |
| 4JHW | 0 | 0 | 0 |

## F.3 NUMBER OF UNIQUE SUCCESSES WITH ADDITIONAL ALL-ATOM CONSTRAINT

Table 11: Number of unique successes with all-atom constraint (out of 1,000 structures) generated by Genie 2, RFDiffusion, and FrameFlow on each single-motif scaffolding task.

| Name | Genie 2 | RFDiffusion | FrameFlow |
|------|---------|-------------|-----------|
| 6E6R_long | 323 | **373** | 103 |
| 6EXZ_long | 312 | 167 | **393** |
| 6E6R_med | **213** | 147 | 92 |
| 1YCR | 123 | 7 | **133** |
| 5TRV_long | **16** | 9 | **16** |
| 6EXZ_med | 52 | 25 | **101** |
| 7MRX_128 | 8 | 14 | **23** |
| 6E6R_short | 24 | 20 | **25** |
| 5TRV_med | **5** | 2 | **5** |
| 7MRX_85 | 14 | 9 | **18** |
| 3IXT | **14** | 3 | 8 |
| 5TPN | **8** | 5 | 6 |
| 7MRX_60 | **4** | 1 | 1 |
| 1QJG | 0 | 1 | **13** |
| 5TRV_short | **2** | 1 | 1 |
| 5YUI | **1** | **1** | **1** |
| 4ZYP | 2 | **5** | 4 |
| 6EXZ_short | 2 | 1 | 3 |
| 1PRW | **1** | **1** | **1** |
| 5IUS | **1** | **1** | 0 |
| 1BCF | **1** | **1** | **1** |
| 5WN9 | 1 | 0 | **3** |
| 2KL8 | **1** | **1** | **1** |
| 4JHW | 0 | 0 | 0 |

### F.4  SCATTERPLOT OF SCRMSD VERSUS MOTIF BACKBONE RMSD

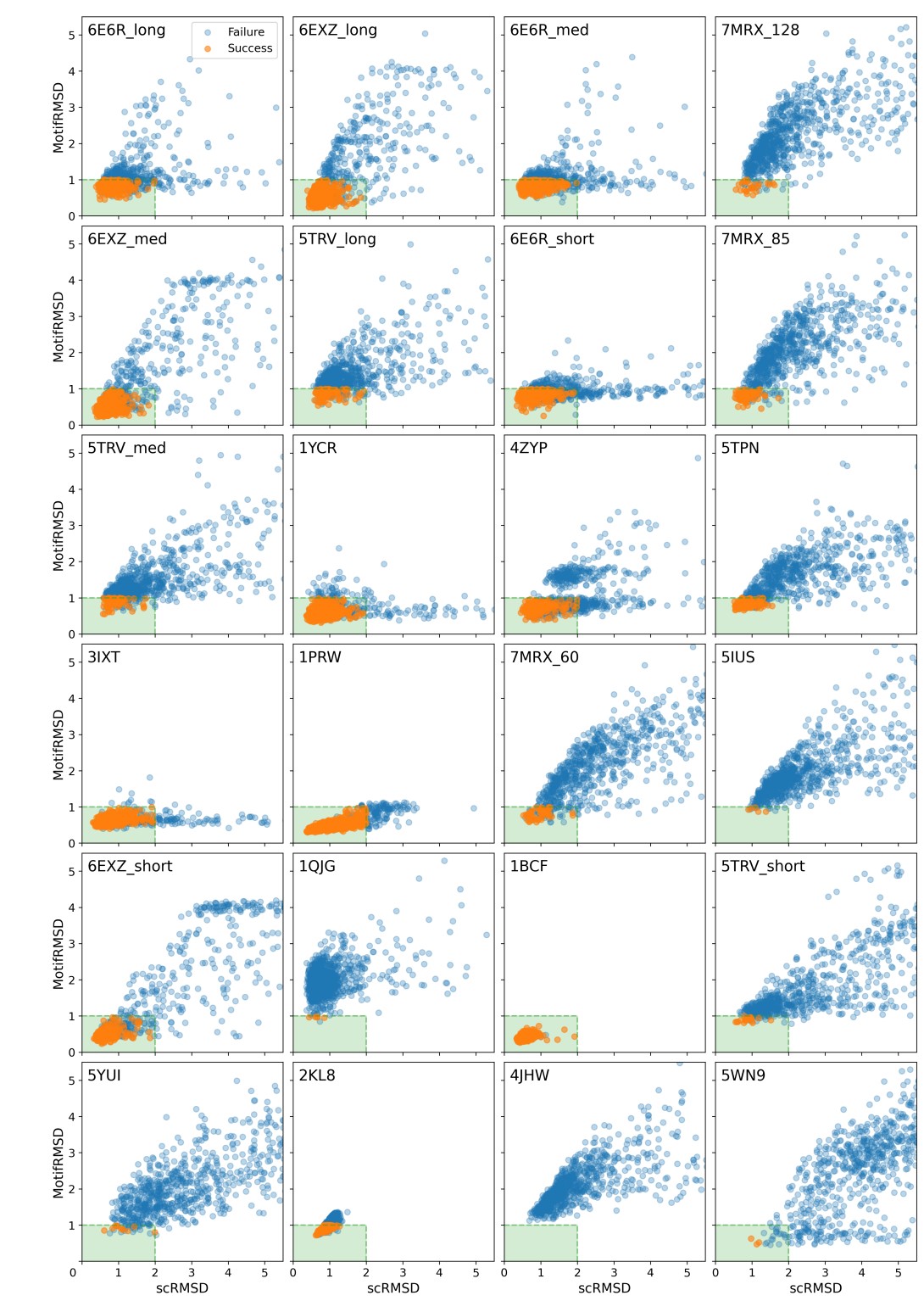

Figure 19: Scatterplot of scRMSD versus motif backbone RMSD by problem, where each point represents one generated structure. The green box denotes the region with scRMSD $\leq$ 2Å and motif backbone RMSD $\leq$ 1Å, which are the two deciding factors of a design's success.

## F.5 ADDITIONAL EXAMPLES OF SUCCESSFUL MOTIF SCAFFOLDING DESIGNS BY GENIE 2

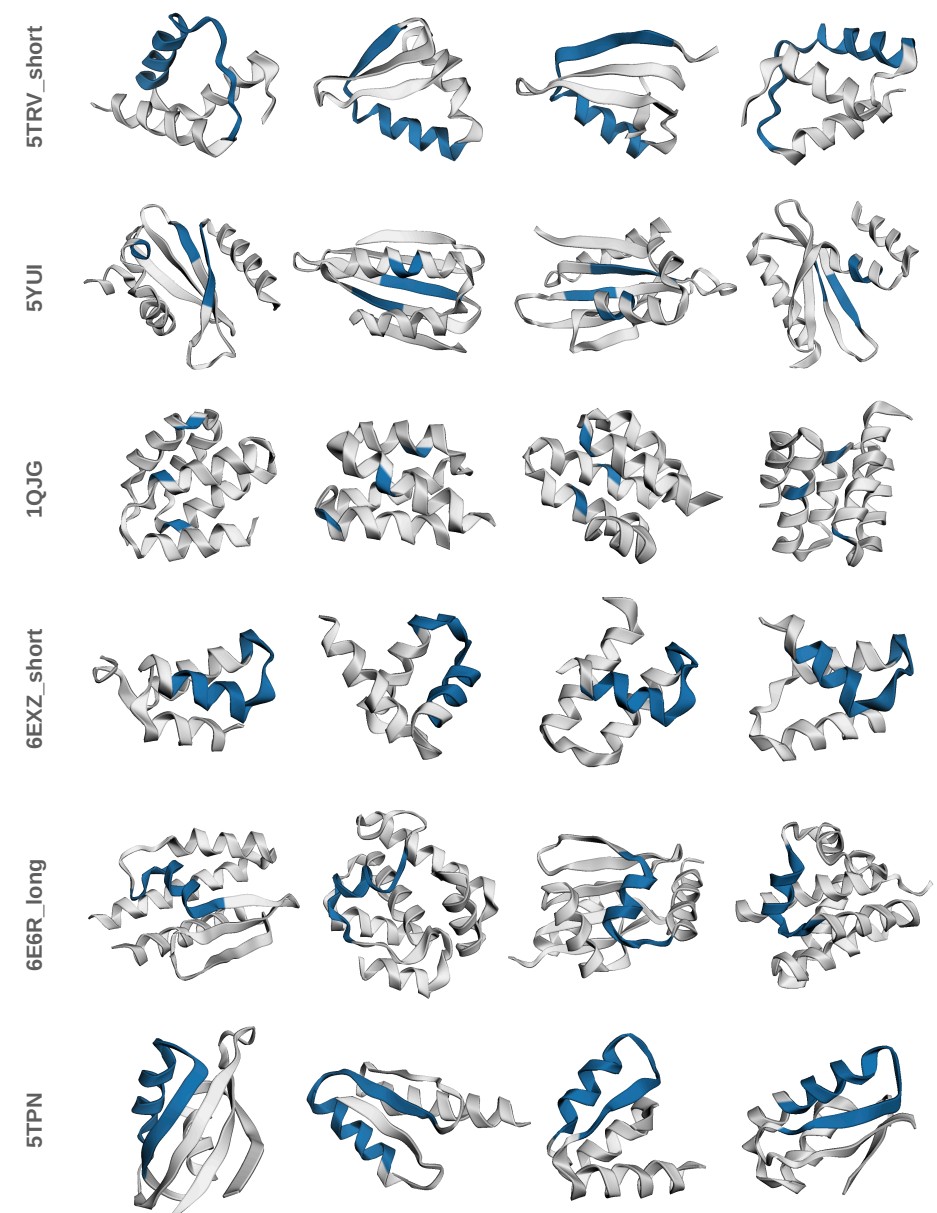

Figure 20: Examples of successfully designed structures by Genie 2 for six single-motif scaffolding tasks. Scaffolds (white) and motifs (blue) are overlaid.

### F.6 COMPARISON WITH CHROMA

Figure 21 compares the performance of single motif scaffolding between Genie 2 and Chroma using the same approach as in the main text, except that we sample only 100 structures for each problem. Genie 2 significantly outperforms Chroma on motif scaffolding, solving more tasks while generating much more diverse solutions at the same time.

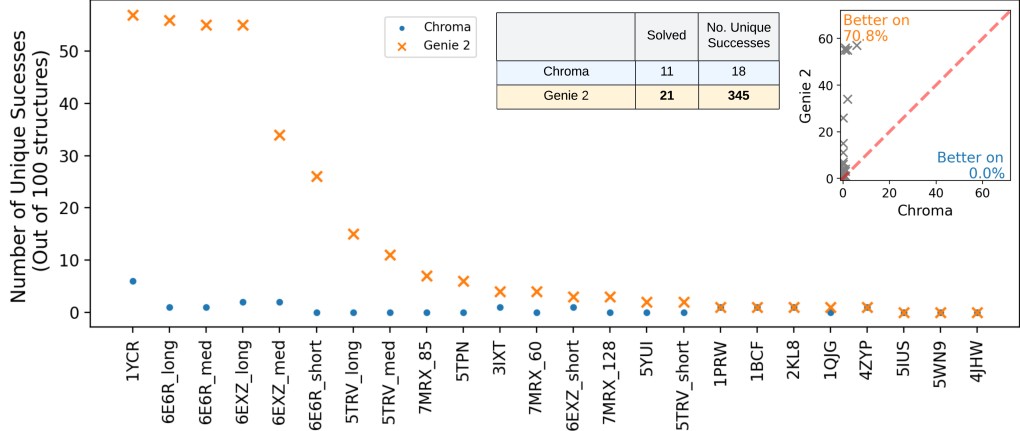

Figure 21: Comparison of Genie 2 and Chroma on single motif scaffolding. Inset (top right) shows a scatter plot of the (unique) success rate of Genie 2 versus Chroma; each point represents a scaffolding task. Summary statistics are shown in table (left).

### F.7    SCAFFOLDING WITHOUT MOTIF SEQUENCE INFORMATION

Motif sequence plays an important role in functional motif scaffolding since functions are generally determined by arrangements of side chain atoms. To investigate the importance of motif sequence for Genie 2, we perform single motif scaffolding without sequence information. We sample structures with Genie 2 by providing only the motif structure information as conditional input. We then follow the same evaluation procedure as described in Section 5.1; note that when inverse folding with ProteinMPNN, motif sequences are kept fixed in an attempt to preserve motif functionality. Figure 22 visualizes the single motif scaffolding performance of Genie 2 with and without motif sequence information as model input. We observe a drop in performance when omitting motif sequences from model input, suggesting that Genie 2 is internalizing sequence information into its latent representations for better protein design.

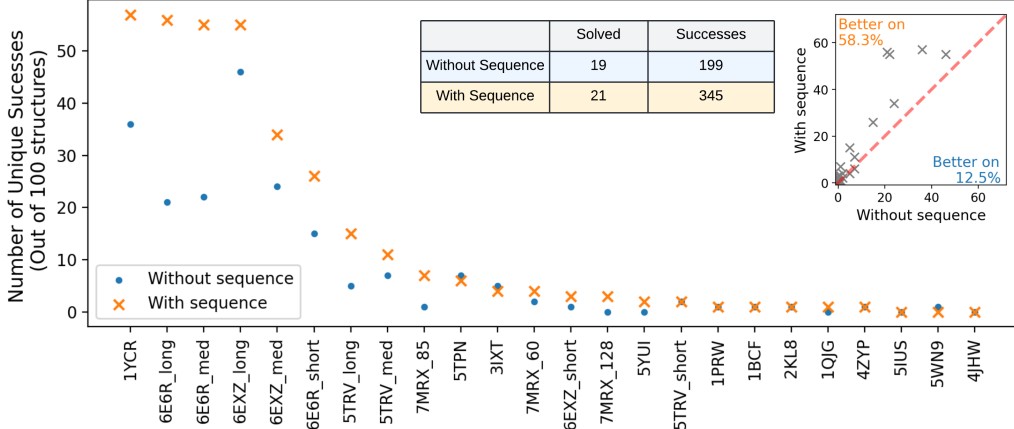

Figure 22: Comparison of Genie 2 on single motif scaffolding with or without motif sequence information as input. Inset (top right) shows a scatter plot of the (unique) success rate; each point represents a scaffolding task. Summary statistics are shown in table (left).

# G   ADDITIONAL RESULTS ON MULTI-MOTIF SCAFFOLDING

## G.1   NUMBER OF UNIQUE SUCCESSES

Table 12: Number of unique successes (out of 1,000 structures) generated by Genie 2 on each multi-motif scaffolding task.

| Name | Number of Unique Successes |
|---|---|
| 3BIK+3BP5 | 17 |
| 1PRW_four | 11 |
| 1PRW_two | 8 |
| 4JHW+5WN9 | 4 |
| 2B5I | 0 |
| 3NTN | 0 |

## G.2   ADDITIONAL EXAMPLES OF SUCCESSFUL DESIGNS BY GENIE 2

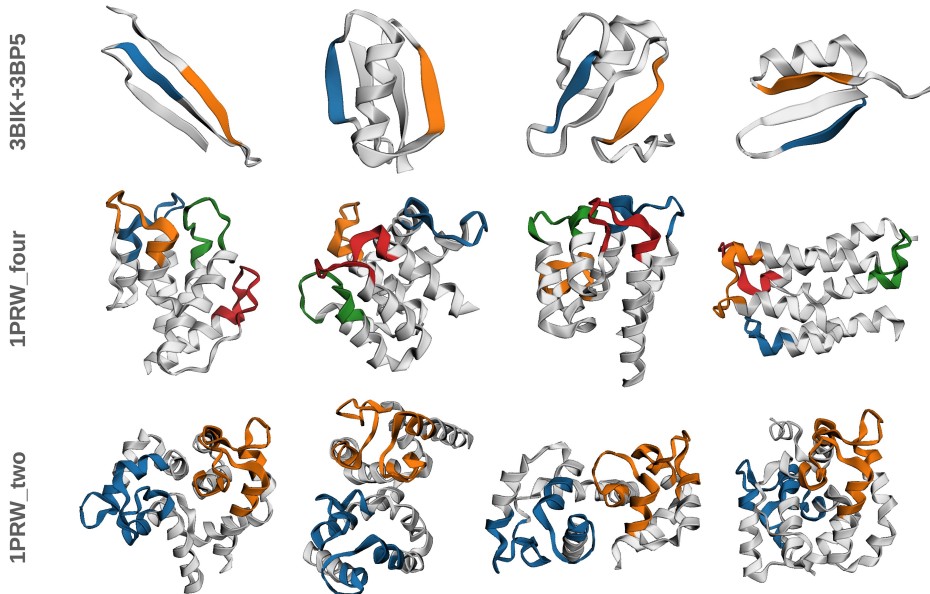

Figure 23: Examples of successfully designed structures by Genie 2 for three multi-motif scaffolding tasks. Scaffolds are in grey and different motifs are colored differently. For 4JHW+4WN9, all four unique successes are shown in Figure 5.

Figure 23 shows the successful designs of three multi-motif scaffolding problems. The designs for 3BIK+3BP5 exhibit diverse secondary structures, including structures containing strands (first), helices (second and fourth), and loops (third). For tasks 1PRW_two and 1PRW_four, the loops of EF-hand motifs that interact with substrates are well exposed to the surface in all designs. The structures are diverse and clearly different from the original 1PRW, with the EF-hand motifs distributed asymmetrically throughout the structure. In the designs of 1PRW_two, the 4-helix bundles are also in different relative orientations compared to the original 1PRW. For example, in the second design, the loops of two bundles face the same side. These diverse and novel designs open the possibility of creating more stable or functional proteins with the desired motifs.

### G.3 CASE STUDIES

In this section, we provide case studies on the two failed multi-motif scaffolding problems, 3NTN and 2B51, to better understand the potential causes of failure. We also experiment on scaffolding multiple $Ca^{2+}$-binding motifs to study the impact of number of sought motifs on generation.

#### G.3.1 PROBLEM 3NTN: SCAFFOLDING TWO 3-HELIX BUNDLES

Problem 3NTN is constructed from a homotrimer (PDB: 3NTN); it consists of a 3-helix bundle (7 residues per helix) that binds to $Cl^-$ ion and a 3-helix bundle (6 residues per helix) that binds to $Ni^{2+}$ ion (Figure 24A). Our goal is to design a monomer that incorporates both binding pockets. Resembling the original protein, we define our multi-motif configuration as shown in Figure 24C, where blue segments correspond to the $Ni^{2+}$ binding site, orange segments correspond to the $Cl^-$ binding site, and gray segments correspond to scaffold segments to be designed.

We visualize the scRMSD vs. maximum motif backbone RMSD for all 1,000 designs generated by Genie 2 in Figure 24B, with the design closest to success shown in Figure 24D. Note that since we are scaffolding multiple motifs with flexible positions and orientations, we compute backbone RMSD between the generated and target structure for each motif separately and report the maximum motif backbone RMSD for the design in Figure 24B. Examining the design closest to success, we observe that Genie 2 is capable of generating the $Cl^-$ binding site but fails to reproduce the $Ni^{2+}$ binding site. In addition, the pAE of the design is 7.29, slightly over our success threshold of 5. We suspect that the predefined multi-motif configuration does not provide sufficient flexibility (*i.e.,* too few residues in the connecting scaffold segments) for the model to generate viable solutions.

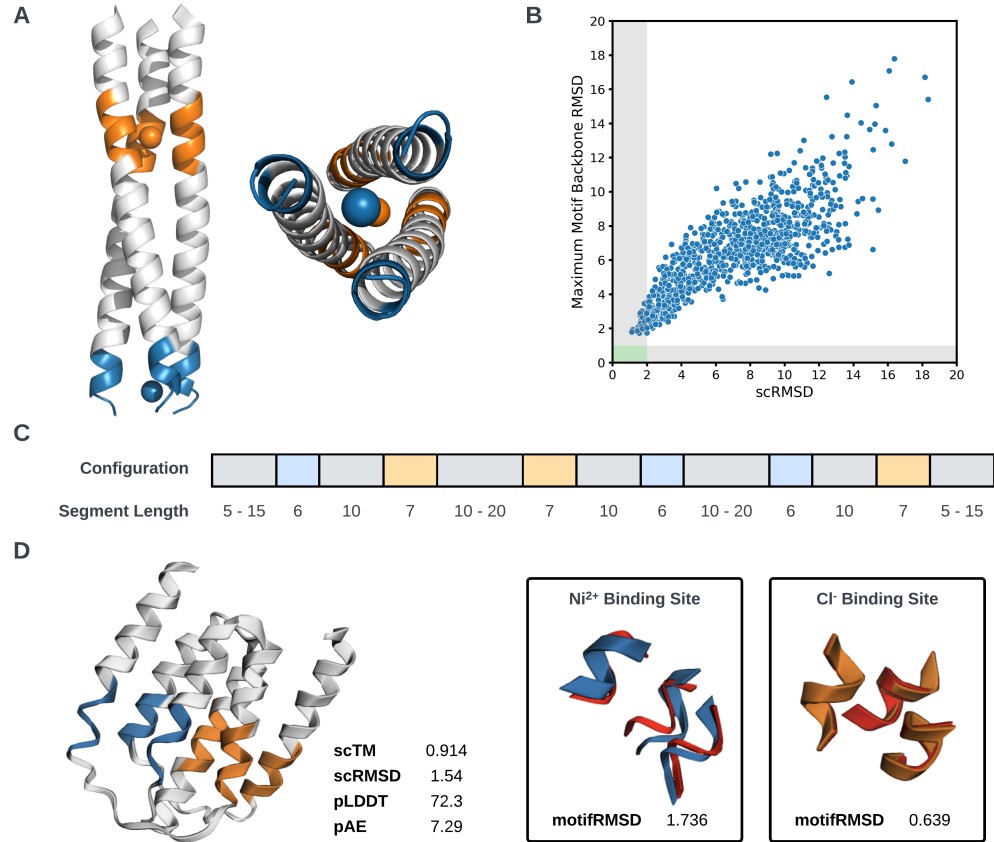

Figure 24: Analysis of the multi-motif scaffolding problem 3NTN. **(A)** Visualization of PDB structure 3NTN. **(B)** Scatter plot of scRMSD vs. maximum motif backbone RMSD across 1,000 designs generated by Genie 2, with success region shown in light green. **(C)** Multi-motif scaffolding configuration for problem 3NTN. **(D)** Closest design to success, with individual binding sites superposed over targets (red).

### G.3.2 PROBLEM 2B5I: SCAFFOLDING TWO PROTEIN BINDING SITES

For problem 2B5I, the goal is to design a protein that binds to both IL-2 receptor $\beta\gamma_c$ heterodimer (IL-2R$\beta\gamma_c$) and IL-2R$\alpha$ (or CD25), thus stabilizing variants of IL-2 in both binding sites (Ren et al., 2022). To construct this multi-motif scaffolding task, we extracted motifs from IL-2 protein (PDB: 2B5I), which forms a complex with IL-2R$\beta\gamma_c$ and IL-2R$\alpha$ (Wang et al., 2005). Figure 25A visualizes these motifs and their interactions with IL-2R$\beta\gamma_c$ and IL-2R$\alpha$, and Figure 25C shows the multi-motif configuration for this problem.

Similar to the previous case, we use a scatter plot to visualize scRMSD vs. maximum motif backbone RMSD for all 1,000 designs generated by Genie 2 in Figure 25B, with the design closest to success shown in Figure 25D. We observe the best design is very close to *in silico* success, with pAE and maximum motif backbone RMSD slightly over the success threshold.

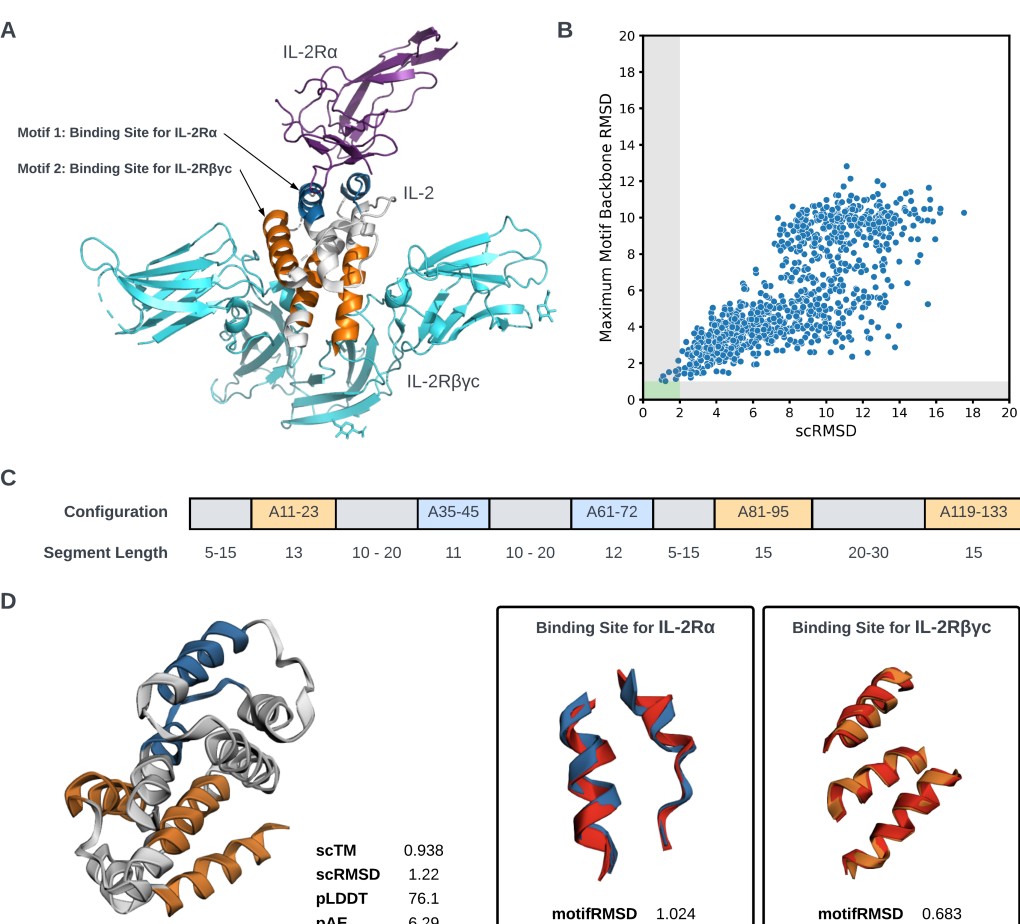

Figure 25: Analysis of multi-motif scaffolding problem 2B5I. **(A)** Visualization of PDB structure 2B5I. **(B)** Scatter plot of scRMSD vs. maximum motif backbone RMSD across 1,000 designs generated by Genie 2, with success region shown in light green. **(C)** Multi-motif scaffolding configuration for problem 2B5I. **(D)** Closest design to success, with individual binding sites superposed over targets (red).

### G.3.3 SCAFFOLDING MULTIPLE EF-HAND CA$^{2+}$-BINDING MOTIFS

To better understand how the number of motifs affects model performance, we perform a controlled study derived from problem 1PRW_four, whose goal is to design a protein that binds with multiple Ca$^{2+}$ ions. We extract an EF-hand (helix-loop-helix) Ca$^{2+}$-binding motif (Figure 26A) from Calmodulin (PDB: 1PRW) and use Genie 2 to design proteins containing increasing numbers of this EF-hand Ca$^{2+}$-binding motif (from 1 to 5). For each settings, we sample 1,000 structures and follow the same evaluation procedure as in the main text. Figure 26C visualizes the multi-motif configuration for this set of problems.

Figure 26B shows Genie 2's performance (in terms of number of successes and unique successes) as the number of motifs increases. Here, the success criteria is same as in the main text, requiring scRMSD $< 2$, pLDDT $> 70$, pAE $< 5$ and all motif backbone RMSD $< 1$. As the number of motifs increases, multi-motif scaffolding performance drops since it is hard to find a viable solution that satisfies all motif constraints; however, the number of unique successes increases (bounded by the number of successes) as the number of motifs increases, possibly because the increase in sequence length provides a larger search space for solutions.

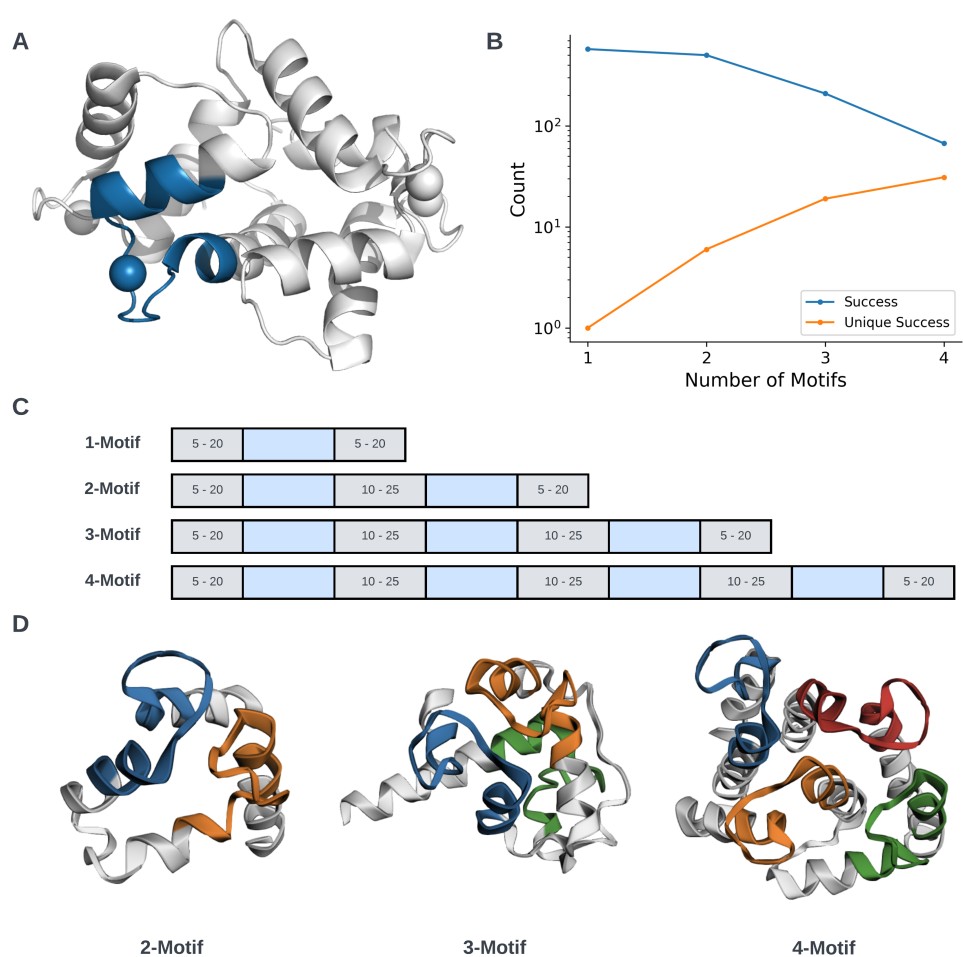

Figure 26: Analysis on scaffolding multiple EF-hand Ca$^{2+}$-binding motifs. **(A)** Visualization of 1PRW (PDB: 1PRW) with the Ca$^{2+}$-binding motif overlaid. **(B)** Number of successes and unique successes as a function of number of motifs. **(C)** Multi-motif configurations for the set of multi-motif scaffolding problems. **(D)** Examples of successful designs.

### G.4 COMPARISON WITH RFDIFFUSION

While RFDiffusion does not provide built-in functionality to scaffold multiple motifs with flexible relative positions and orientations, it is possible to adapt it for multi-motif scaffolding tasks by randomly sampling relative positions and orientations between motifs and scaffolding these motifs as a whole with RFDiffusion. We adopt this approach here and compare the performance of RFDiffusion on multi-motif scaffolding with Genie 2, using the set of multi-motif scaffolding problems detailed in Appendix C.

For two double-motif scaffolding problems (3BIK+3BP5 and 4JHW+5WN9), where each motif consists of a single segment, we sampled 100 double-motif configurations by randomly rotating the motifs and setting the distance between two motif centers to be $r$, where $r \sim \mathcal{U}(5, 15)$ and $r \sim \mathcal{U}(10, 25)$, for problems 3BIK+3BP5 and 4JHW+5WN9, respectively. These parameters are carefully selected based on motif sizes. Any configuration with clashes (*i.e.,* distance between two atoms from different motifs <2Å) between motifs is rejected. We then generated 10 designs for each configuration, summing to 1,000 designs in total. For the other four problems, we generate 1,000 multi-motif configurations by randomly rotating all the motifs and sampling motif centers from a 50Å cube such that all motif centers are at least 10Å distance from one another. As before, configuration with clashes between motifs are rejected. We then generated 1 design for each configuration, summing to 1,000 designs in total.

Table 13 compares the performance of RFDiffusion and Genie 2 on multi-motif scaffolding tasks. In terms of unique successes, Genie 2 outperforms RFDiffusion on 3 out of 4 solved problems, while RFDiffusion performs better on problem 1PRW_two. Our hypothesis is that while setting random relative positions and orientations between motifs creates many invalid motif configurations, it explicitly encourages diversity; in contrast, for Genie 2, diversity relies on the reverse diffusion process, resulting in differing probabilities with which each viable solution is sampled. This could explain why Genie 2 generates more successes than RFDiffusion on problem 1PRW_two (since it implicitly reasons over relative motif positions and orientations) but fewer unique successes (since some viable solutions have a high probability of being sampled in the sampling stage). It is worth noting that while RFDiffusion is able to produce viable designs for double-motif scaffolding problems, it struggles to do so for problems involving more motifs (for example, problem 1PRW_four, which involves four EF-hand $Ca^{2+}$-binding motifs) due to the exponentially increasing number of possible multi-motif configurations.

Table 13: Number of successes and unique successes (out of 1,000 structures) generated by Genie 2 and RFDiffusion on each multi-motif scaffolding task.

| PROBLEM | SUCCESSES | | UNIQUE SUCCESSES | |
|---|---|---|---|---|
| | GENIE 2 | RFDIFFUSION | GENIE 2 | RFDIFFUSION |
| 3BIK+3BP5 | **172** | 12 | **17** | 6 |
| 1PRW_four | **14** | 0 | **11** | 0 |
| 1PRW_two | **324** | 34 | 8 | **33** |
| 4JHW+5WN9 | **4** | 2 | **4** | 2 |
| 2B5I | 0 | 0 | 0 | 0 |
| 3NTN | 0 | 0 | 0 | 0 |

## H    SAMPLING TIME

In this section, we compare the generation times of Genie 2, RFDiffusion, FrameFlow, and Chroma at different lengths. We use a single A6000 GPU (48GB memory) and average the inference time of a single sample over 10 runs. For RFDiffusion, we use the self-conditioning sampler and exclude pLDDT and amino acid prediction for fair comparison. For Chroma, we use unconditional monomer sampling and exclude amino acid prediction. We use the simple profiler from PyTorch Lightning (Falcon and The PyTorch Lightning team, 2019) to profile inference function calls of FrameFlow and Genie. For Proteus, we used their logger output with the self-consistency pipeline disabled.

Table 14: Sampling time of different methods for proteins of different lengths. For Genie 2, we use the built-in PyTorch compilation function to speed up the sampling process.

| Method | Parameters | Timesteps | Length | | | | | |
|--------|-----------|-----------|--------|--------|--------|--------|--------|--------|
| | | | 50 | 100 | 200 | 300 | 400 | 500 |
| Genie 2 | 15.7M | 1000 | 15.1 | 18.8 | 41.3 | 83.0 | 152 | 223 |
| | | 500 | 4.04 | 7.54 | 18.90 | 39.5 | 66.4 | 103 |
| | | 200 | 1.76 | 3.01 | 7.66 | 15.9 | 26.11 | 41.3 |
| | | 100 | **0.79** | **1.54** | **3.80** | **8.02** | 13.2 | 20.8 |
| RFDiffusion | 59.8M | 50 | 18.7 | 21.4 | 41.2 | 80.1 | 137 | 214 |
| FrameFlow | 17.4M | 100 | 3.45 | 4.33 | 6.50 | 9.84 | 13.8 | 18.5 |
| Chroma | 18.5M | 500 | 22.0 | 22.6 | 29.0 | 35.4 | 41.8 | 48.5 |
| Proteus | 19.8M | 100 | 5.10 | 5.23 | 6.40 | 9.07 | **12.7** | **17.5** |

