# OpenReview forum: "Out of Many, One: Designing and Scaffolding Proteins at the Scale of the Structural Universe with Genie 2"
_ICLR.cc/2025/Conference — Submitted to ICLR 2025_

### Official Review · Reviewer_4r8p · 2024-11-04

**Soundness:** 3
**Presentation:** 3
**Contribution:** 2
**Rating:** 5
**Confidence:** 3

**Summary:**

This work builds on Genie 1 by adding motif scaffolding, enabling the generation of protein scaffolds that support specified motifs - an important step toward the ultimate goal of designing functional proteins.

The authors developed a simple yet effective method to encode motif information as node and pair conditions, enabling the generation of protein structures that closely adhere to targeted motif cores. They also extend the single-motif problem to multi-motif generation, exploring whether the model can generate proteins with independent functions, pushing the boundaries of creating more complex and versatile protein machines. Additionally, they found that augmenting training with AFDB clusters and applying conditional training benefits both motif scaffolding and unconditional structure generation, with benchmarks against strong SOTA models, including RFDiffusion, FrameFlow, and Chroma.

In summary, this work provides an interesting perspective on protein motif design and demonstrates the improvements of Genie 2 in protein structure generation.

**Strengths:**

**[Clarity and Quality]**

- The manuscript is clearly structured and easy to follow, with a comprehensive background review covering sequence, structure, and co-design, as well as motif scaffolding. This allows readers to understand the main concepts without needing to refer to external sources.
- The authors conducted sufficient experiments to evaluate empirical performance of Genie 2 on both the unconditional generation and scaffolding, comparing it with strong SOTA baselines such as RFDiffusion and FrameFlow.

**[Originality and Significance]**

- The authors extend motif scaffolding to design multiple independent motifs, supported by a benchmark problem set, which could benefit protein generation in more complex protein machinery.
- By encoding motif conditions as node and pair embeddings, the authors provide an SE3-invariant and flexible approach for representing motif information.

**Weaknesses:**

**[Quality and Clarity]**

- The current version lacks in-depth studies on key tuning parameters (e.g., sampling temperature) and critical design factors (e.g., conditional training) (see Q1/Q3).
- Some important baselines are missing (see Q8/Q9).
- Some analyses and statements need further evaluation (see Q5/Q6/Q7).
- Certain statements are vague and would benefit from clarification (see Q2/Q12).

**[Significance and Originality]**

While the independent multi-motif design presents an interesting task, the current work faces limitations that affect its impact:

- The primary improvement of Genie 2 over current SOTA appears to be in diversity (including diverse solutions in motif scaffolding). However, it remains unclear whether this improvement is due to training with AFDB or low-temperature sampling (see Q3).
- Training on AFDB is a logical approach that has already been applied in other protein design models (e.g., Bilingual Language Model for Protein Sequence and Structure, Michael Heinzinger et al., bioRxiv, https://doi.org/10.1101/2023.07.23.550085).
- Encoding protein structure as SE3-invariant node and pair features is also an established approach (e.g., AlphaFold 2 encodes template information as node and pair). Although this work shows that SE3 invariance is effective for motifs encoding, it does not demonstrate broader impact on this design choice.

**Questions:**

1. Table 3: The authors demonstrate the effect of conditional training ratio on small-scale, “unconverged” models. While this limitation is understandable due to time constraints, the conclusion that conditional training universally enhances performance remains speculative. More comprehensive results during the rebuttal period would strengthen the findings.
2. Low-temperature sampling (line 323): The authors state that “low temperature sampling yields better results.” Could they specify which metric(s) this “better” refers to, as it’s unclear from the context?
3. Temperature sensitivity: The authors used different temperatures (γ) for unconditional generation and motif-scaffolding, highlighting the importance of tuning this parameter. Could they include the default temperatures used for each model in the main text and clarify the impact of varying temperatures on each task? Does this adjustment impose trade-offs, and how sensitive is the model to this factor?
4. Training data balance: Is it reasonable to balance experimental structure data (PDB) and synthetic data (AFDB) for training? Are the results significantly biased towards synthetic data?
5. Secondary structure distribution (section 4.2): When evaluating secondary structure distribution, comparing to AFDB might be problematic as 1) not all baselines were trained on AFDB, and 2) AFDB may differ in structural distribution from real-world PDB data.
6. Secondary structure distribution (section 4.2): Contrary to the authors' suggestion in line 318, it seems Chroma, Proteus, and RFDiffusion appear to generate more beta strands (top left) than AFDB. Could the authors clarify their statement about underrepresented beta strands and loop elements?
7. Low-temperature sampling for AFDB (line 352): The lower designability in AFDB might explain low designability at normal temperatures for Genie 2, but it doesn’t directly explain why low-temperature sampling would mitigate this. Could the authors provide additional insights?
8. Baseline model: The authors mention the Twisted Diffusion Sampler in related work but do not include it as a baseline. Could they clarify this choice?
9. Baseline model Genie 1: Could the authors include baseline performance results for Genie 1 in unconditional generation?
10. Multi-motif scaffolding benchmark curation: This benchmark includes only proteins with <200 amino acids. Have the authors explored more complex cases with longer sequences, or are there specific challenges in curating such datasets?
11. Multi-chain generation: Although Genie 2 accepts “chain index” as an input feature, it does not support multi-chain generation with motif-scaffolding. What limits this implementation?
12. Clarifications on some claims:
    - The authors claim Genie 2 “better captures protein structure space” and covers a “larger and more diverse protein structure space.” However, this aspect has not been directly evaluated. Current results only show higher diversity, not necessarily “broader structure space coverage” compared to Genie 1, especially given the smaller sample size examined (1,035 structures, line 309).
    - In Equation (5), the authors claim that “motifs are enforced as a soft constraint, ensuring responsiveness to motif specifications while designing the protein as a whole.” However, the current loss does not distinguish explicitly between motifs and scaffolds (e.g., no separate regularization for motif losses). The authors might want to suggest the benefits of jointly learning the motif and scaffolds coordinates and further clarification could strengthen the interpretation.

---

> ### Author Response · Authors · 2024-11-23
> **Response by Authors (Part 1/3)**
>
> We would like to thank the reviewer for the constructive and thoughtful feedback. Please see our responses below:
>
> **Weakness on Quality and Clarity**: Please refer to our responses to questions below.
>
> **Weakness on Significance and Originality**: We believe that our improved diversity results from the combined effect of training on AFDB and sampling at a lower noise scale. Training on AFDB allows us to leverage a much larger structural space than that of the PDB and results in a better protein diffusion model that can tackle new protein design tasks unaddressed by existing methods. Training a leading protein diffusion model on the whole of the clustered AFDB database, while apparently conceptually simple, in fact requires overcoming substantial engineering challenges as well as making critical design choices. This is supported by the fact that while AFDB has been available since 2021, Genie 2 is the first to learn to model the data distribution of AFDB structures. Some challenges of modeling the data distribution of AFDB structures include structural quality (since they are not experimentally validated), existence of multi-domain structures and the scale of AFDB (over 200 millions structures). Indeed, soon after Genie 2 was preprinted, FoldFlow-2 came out as a contemporary method that attempted to use AFDB, but found that training on AFDB actually  hampered model performance. An innovation of this work is the use of low noise-scale sampling in combination with AFDB, which we believe contracts average protein radius and encourages the generation of more compact (and thus more designable) structures (refer to Appendix E.3 for more discussion on low sampling noise scale). By combining training on AFDB with low noise-scale sampling, Genie 2 is able to leverage the structural diversity of AFDB while retaining high quality generation capability, enabling successful utilization of AFDB.
>
> **Question 1**: We agree that due to undertraining, the conclusion that higher conditioning task ratios lead to better model performance is speculative and we will update our manuscript to reflect this uncertainty. Our hypothesis is that a higher conditioning task ratio may help speed up training, but model performance may well be unaffected upon convergence since the underlying data distribution is the same.
>
> **Question 2**: The primary metric that we are considering is diversity, which computes the percentage of unique AND designable clusters (given a fixed number of samples). This better reflects model performance than designability alone, which can be inflated when the model repeatedly generates the same designable structures (reflecting overfitting or mode collapse).
>
> **Question 3 & 7**: We now include an additional ablation study on the effect of sampling noise scale in Appendix E.3. We also provide further details on how sampling noise scale is set for other methods in Appendix A.5. We find sampling noise scale to be a model-specific hyperparameter, and for fair comparisons with other methods, we use the pretrained weights and default hyperparameters provided in their GitHub repositories, which should provide the best generative performance. Upon revising the manuscript for the final version, we will include the default temperatures used for each model in the main text.
>
> **Question 4**: Yes it is a reasonable suggestion to balance experimental and synthetic structures during training (this is done by AlphaFold3 for example). The primary reason that we only use AFDB filtered clusters is that these clusters are more structurally diverse and are expected to cover existing structures within the PDB. Training on both PDB and AFDB structures could introduce bias towards the intersection of two sets of structures and would introduce another hyperparameter to tune. It is something we will consider for future work however.
>
> Regarding the bias on synthetic dataset, it is true that by training on filtered AFDB structures, the model is learning to model the data distribution of AFDB. However, from the perspective of structural diversity, AFDB is a super set of the PDB monomers and Genie 2’s expanded model capacity appears able to capture both (Appendix E.2). We believe that this is (indirectly) demonstrated by the fact that Genie 2 solves problems that methods trained on the PDB alone are unable to solve, suggesting that Genie 2 can tap into a larger structural space.

---

> > ### Author Response · Authors · 2024-11-23
> > **Response by Authors (Part 3/3)**
> >
> > **Question 12, part 1**: We argue that increased diversity implies coverage of a larger structure space, as the diversity of a fixed set of proteins effectively quantifies the size of the structure space covered by them. It is true that if our sampling procedure is biased, it may be that certain regions of structure space are more or less frequently sampled than other regions, and a fixed sample size of generated structures may underestimate or overestimate the size of implied structure space. Our claim was meant to be heuristic, and we will clarify this in the main text.
> >
> > We have done an additional study that computes the TM score of generated structures vs. structures in the PDB and AFDB (Appendix E.2) and found that for many designs, they find closer hits to the AFDB, suggesting that the model is leveraging its expanded training set when designing new proteins, and this too implies capturing a larger structure space than a model trained strictly on the PDB.
> >
> > **Question 12, part 2**: We equally weigh motifs and scaffolds during training since overweighting the motif or additional regularization loss on motif could potentially skew the learning signal towards only memorizing motif structures instead of learning the data distribution of protein structures. Thus, when designing the training process of Genie 2, we decide not to introduce an additional hyperparameter that controls the weight on motif loss. The key benefit of jointly learning motif and scaffold coordinates is that the model implicitly reasons the validity of the protein as a whole, resulting in generation of more designable structures.

---

> > > ### Comment · Reviewer_4r8p · 2024-11-26
> > >
> > > I appreciate the additional details and extended results provided in the updated manuscript - they have addressed my questions. I have adjusted my score.

---

> > > > ### Author Response · Authors · 2024-11-29
> > > > **Thank you for your response**
> > > >
> > > > We appreciate the time and effort you have put in reviewing our work. We see that your questions have been addressed. This usually implies that the paper would be above the bar for acceptance. We would like to ask if it is possible to provide further clarification for how the manuscript could have been improved above a score of 5 (below acceptance threshold). Here we note that Genie 2 is the current SOTA method for unconditional generation and motif scaffolding. In particular, in the revised version of our manuscript we have demonstrated that:
> > > > - Genie 2 outperforms all methods in consideration including RFDiffusion, Chroma, CarbonNovo and MultiFlow by achieving comparable designability (0.96) but higher diversity and novelty (Appendix E.7; closest competitor is RFDiffusion which achieves F1: 0.76, diversity: 0.63, and novelty: 0.26 vs. Genie 2 which achieves F1: 0.93, diversity: 0.91, and novelty: 0.41). We note that the deltas are rather substantial, and so this is not merely an incremental claim of achieving SOTA.
> > > > - These results are achieved even with much fewer diffusion steps and thus higher sampling efficiency (Appendix B.2, H). Together these results establish a much stronger foundation than what is currently available for future protein design methods.
> > > > - Genie 2 significantly outperforms existing methods on generating longer proteins (Appendix E.4), generating significantly more diverse designable clusters than existing methods for proteins with more than 300 residues (2.5-fold higher diversity compared to Proteus on generating proteins with 500 residues)
> > > > - Genie 2 outperforms all methods in consideration for motif scaffolding (Figure 4, Appendix F.6). Compared to leading motif scaffolding methods including RFDiffusion, Chroma and FrameFlow-Amortized, Genie 2 is capable of solving more motif scaffolding problems with more diversity (closest competitor is FrameFlow-Amortized which solves 22 problems with a total of 1,099 unique solutions vs. Genie 2 which solves 23 problems with a total of 1,445 unique solutions).
> > > > - Genie 2 also provides additional supports for multi-motif scaffolding (with detailed analysis on Appendix G.3) by implicitly reasoning over the relative positions and orientations among motifs, which is a capability not supported by leading protein diffusion models such as RFDiffusion. While one can combine RFDiffusion with randomly sampled relative positions and orientations between motifs, the success rate and thus sampling efficiency is much lower than Genie 2 (Appendix G.4).
> > > > - Lastly, Genie 2 is the first protein design model to successfully train on the scale of the known protein structural universe. Combined with low scale sampling (Appendix E.3), Genie 2 is able to leverage the diversity of the AlphaFold database while maintaining high designability during generation.

---

> > > > > ### Comment · Reviewer_4r8p · 2024-12-02
> > > > >
> > > > > Thanks for the further note. I raised my score as the authors provided more discussion and results that addressed my questions (mostly technical), improving the overall clarity and quality of the manuscript. However, I did not raise it further due to concerns about the overall contribution and significance of the work:
> > > > >
> > > > > - The main architectural contributions are limited. Most of the model modules are inherited from AlphaFold2 and follow common practices in the field. Adding template conditioning into node and pair representations also closely resembles the  template input in AlphaFold2.
> > > > >
> > > > > - The improvements in diversity, the key metric suggested by the authors, are result from combining training on a large synthetic AFDB dataset (which is diverse but may include low-quality samples) with low-temperature/low-noise-scale sampling (trading diversity for quality to counterbalance). Since both the dataset and sampling techniques are available, this combination seems straightforward rather than sufficiently innovative.
> > > > >
> > > > > These factors lead me to question whether these contributions are strong technical advancements that are sufficient to warrant publication.

---

> ### Author Response · Authors · 2024-11-23
> **Response by Authors (Part 2/3)**
>
> **Question 5 & 6**: We now visualize the secondary structure distribution of the PDB (similarly to how we do this for AFDB, using 1,000 randomly selected structures) in Figure 17, which closely resembles that of our filtered AFDB dataset. Indeed, when comparing to AFDB, PDB contains a slightly higher content of stranded structures and this could potentially explain why Chroma, Proteus, and RFDiffusion generate slightly more beta strands since they are trained to capture the PDB data distribution while Genie 2 is trained to capture the AFDB data distribution.
>
> We do note that the PDB does not necessarily reflect the natural distribution of proteins across life. The PDB is highly biased in its distribution towards experimentally-tractable structures, model organisms, and biomedically relevant proteins. We expect the fold distribution of the PDB to look quite different from overall protein structure space. The AFDB may indeed be more representative (given that it covers a large fraction of all sequenced organisms), and in particular, we use a clustered version of the AFDB which at least ensures that all protein folds are roughly equally represented.
>
> Regarding the argument for underrepresented beta sheets and loop elements, we meant that existing models do not model structures with less than 50% secondary structure content (regardless of type). For example, RFDiffusion explicitly filters out structures with less than 50% secondary structure when generating the training dataset, possibly because these structures are more flexible and thus have less well-determined structures. This implies that the self-consistency pipeline might not be a good in silico evaluation pipeline for these flexible structures. This could also explain why Genie 2 performs poorly when normal sampling noise scale (refer to Appendix E.3 for more details). As shown in Figure 2B, Genie 2 is capable of generating structures with low secondary structure contents. However, the flexibility of these structures would render the existing self-consistency metrics ineffective, since there is no one correct solution for these flexible structures.
>
> **Question 8**: The reason we did not include Twisted Diffusion Sampler as a baseline is due to it being a guidance-based approach, which does not take motif sequence and structure as input to the model but uses guidance to steer the generation process. In contrast, other methods like Genie 2, RFDiffusion, and FrameFlow-Amortized take motif sequence and structure information as model input, putting TDS at a comparative disadvantage. We do not expect TDS to outperform Genie 2 and FrameFlow-Amortized since FrameFlow-Amortized outperforms TDS on motif scaffolding and thus for our comparison, we select both RFDiffusion and FrameFlow-Amortized to provide a fair assessment of Genie 2 against existing SOTA methods.
>
> **Question 9**: We now include comparisons on unconditional generation performance between Genie and Genie 2 in Appendix E.1.
>
> **Question 10**: When curating the set of multi-motif scaffolding problems, we went through the literature in search for applications of multi-motif scaffolding; however, existing applications are limited and our dataset comprises effectively all known multi-motif scaffolding problems from the literature. The key reason that our benchmark problems (as well as recent motif scaffolding problems) focus on proteins with at most 200 residues is that for functional protein design, we generally prefer shorter proteins with the target functional motif, mainly because shorter proteins are easier to synthesize and test experimentally and have higher specificity. Practically all successful protein designs are in this length range. Hence, when assessing the performance of Genie 2, we only profile generation of longer proteins (more than 300 residues) on the unconditional generation task and not on motif scaffolding.
>
> **Question 11**: We are actively developing support for multimer generation, including curation of a larger multimer dataset beyond existing PDB multimer structures and designing reliable and efficient pipelines for assessing multimer quality. We do not expect to be able to complete this work in time for this manuscript but we will integrate it into a future version of Genie.

---

### Official Review · Reviewer_pv9F · 2024-11-04

**Soundness:** 2
**Presentation:** 2
**Contribution:** 2
**Rating:** 5
**Confidence:** 3

**Summary:**

This paper presents Genie 2, an improved protein diffusion model that extends the capabilities of its predecessor Genie. The key innovation is the addition of multi-motif scaffolding capabilities that allow designing proteins with multiple functional sites without specifying their relative positions and orientations. The model uses SE(3)-equivariant attention for the reverse diffusion process and introduces a novel motif representation approach. The authors demonstrate state-of-the-art performance in both unconditional and conditional protein generation tasks, particularly in terms of designability, diversity, and novelty.

**Strengths:**

1. Technical Innovation: The paper introduces a clever representation for multi-motif scaffolding that uses distance matrices to specify intra-motif but not inter-motif distances, allowing flexible placement of functional sites.
2. Comprehensive Evaluation: The authors provide thorough comparisons with existing methods like RFDiffusion, FrameFlow, and Chroma across multiple metrics on unconditional protein generation and single-motif scaffolding.
3. Data Innovation: The use of AlphaFold database for training data augmentation is a novel approach that expands the structural space beyond experimentally determined structures.

**Weaknesses:**

1. Training Constraints: The model is limited to training on sequences up to 256 residues in length, though it can generate longer sequences. The implications of this limitation could be better explored.
2. Multi-motif Benchmark: The benchmark set of only 6 multi-motif scaffolding problems is too small to draw statistically significant conclusions about the model's capabilities. A robust benchmark should include at least dozens of diverse cases to properly evaluate performance.
3. Multi-motif Baselines: Although previous works like RFDiffusion were not specifically designed for multi-motif scaffolding, they could still address such problems by randomly sampling relative positions and orientations between motifs. The authors should have included comparisons with these existing methods to demonstrate the advantages of their approach over such a baseline strategy.

**Questions:**

Similar to what was mentioned in the weaknesses section.

---

> ### Author Response · Authors · 2024-11-23
>
> We would like to thank the reviewer for the constructive and thoughtful feedback. Please see our responses below:
>
> **Weakness 1**: The decision to fix maximum sequence length to 256 residues during training was made based on available computational resources. The major computational bottleneck in the current Genie 2 architecture is the space complexity of the triangular multiplicative update layers, which require $O(N^3)$ memory (where N is the maximum sequence length). We are actively working on reducing the memory usage of Genie 2 through integration of deepspeed optimization strategies to support training on larger proteins.
>
> Since the initial submission of the manuscript, we have also trained Genie 2 for additional epochs and observed substantially improved performance on the generation of longer proteins. The results and analysis are in Appendix E.4, which shows superior performance over all competing methods in generating diverse long proteins.
>
> Given that Genie 2 is currently the SOTA method (with several additional and very recent baselines in the revised Appendix), we would welcome a clarification from the reviewer as to why the method only merits a rating of 3 (“reject, not good enough”), given the importance of the protein design problem and the substantially improved performance of Genie 2 (our SOTA gains are not merely incremental relative to existing methods.) An expanded explanation from the reviewer would help us address any outstanding issues now or in a future version of the manuscript, and would be much appreciated as it can lead us to actionable changes.
>
> **Weakness 2**: When assessing multi-motif scaffolding performance, we aimed to pursue realistic design tasks known from the biological literature with proven utility, instead of artificial tasks that may inflate (or deflate) the performance of the evaluated methods. Because of the difficult and cutting-edge nature of multi-motif scaffolding, current known examples are limited to the 6 multi-motif scaffolding tasks curated in our evaluation dataset. This problem is too new to enable more systematic benchmarking, although we agree it would be preferred. We hope that future design tasks will enable retrospective benchmarking of Genie 2, but we do not believe that the nascence of the problem should preclude publication of Genie 2, given its demonstrated performance on the 6 tasks and on conventional motif scaffolding. This is further bolstered by the fact that other methods surveyed, for instance Chroma, fail on all 6 multi-motif scaffolding tasks, which suggests that these tasks are providing a signal regarding the performance of protein design methods.
>
> We did consider the possibility of constructing artificial multi-motif scaffolding tasks using the PROSITE motif database [1] as a source of hypothetical motifs. However, PROSITE is a sequence database whose motifs are identified by scanning a predefined set of rules and patterns against sequences. This does not guarantee the formation of functional motifs due to various reasons; for example, multiple segments of a single motif can be structurally dispersed and unlikely to carry out an actual function. Hence, due to the lack of structural motif databases, we decide to curate a set of multi-motif scaffolding tasks based on recent literature.
>
> **Weakness 3**: We now compare the performance of Genie 2 with RFDiffusion on multi-motif scaffolding. Since RFDiffusion does not support flexible motif positions and orientations, we randomly sample positions and orientations among motifs as suggested by the reviewer. We report the new results in Appendix G.4. While RFDiffusion is able to generate viable solutions for some multi-motif scaffolding problems, Genie 2 generally produces more diverse solutions. In addition, Genie 2 is much more sample efficient since it is able to reason over the relative positions and orientations between motifs, while RFDiffusion relies on randomness to obtain viable multi-motif configurations. This implies that as the number of motifs increases, RFDiffusion will be less likely to succeed since the search space for viable multi-motif configurations expands exponentially. This is supported by the observation that Genie 2 solves problem 1PRW_four, which involves four EF-hand $\text{Ca}^{2+}$ binding motifs, while RFDiffusion fails.
>
>
> **Reference**
>
> [1] Sigrist, Christian JA, Edouard De Castro, Lorenzo Cerutti, Béatrice A. Cuche, Nicolas Hulo, Alan Bridge, Lydie Bougueleret, and Ioannis Xenarios. "New and continuing developments at PROSITE." Nucleic acids research 41, no. D1 (2012): D344-D347.

---

> > ### Comment · Reviewer_pv9F · 2024-11-26
> >
> > Thanks for your effort. I raised my score. Good luck!

---

> > > ### Author Response · Authors · 2024-11-29
> > > **Thank you for your response**
> > >
> > > Thank you for your response and increasing your score to 5. We appreciate the time and effort you have put in reviewing our work. We would like to ask if it is possible to provide further clarification for how the manuscript could have been improved above a score of 5 (below acceptance threshold). Here we note that Genie 2 is the current SOTA method for unconditional generation and motif scaffolding. In particular, in the revised version of our manuscript we have demonstrated that:
> > >
> > > - Genie 2 outperforms all methods in consideration including RFDiffusion, Chroma, CarbonNovo and MultiFlow by achieving comparable designability (0.96) but higher diversity and novelty (Appendix E.7; closest competitor is RFDiffusion which achieves F1: 0.76, diversity: 0.63, and novelty: 0.26 vs. Genie 2 which achieves F1: 0.93, diversity: 0.91, and novelty: 0.41). We note that the deltas are rather substantial, and so this is not merely an incremental claim of achieving SOTA.
> > > - These results are achieved even with much fewer diffusion steps and thus higher sampling efficiency (Appendix B.2, H). Together these results establish a much stronger foundation than what is currently available for future protein design methods.
> > > - Genie 2 significantly outperforms existing methods on generating longer proteins (Appendix E.4), generating significantly more diverse designable clusters than existing methods for proteins with more than 300 residues (2.5-fold higher diversity compared to Proteus on generating proteins with 500 residues)
> > > - Genie 2 outperforms all methods in consideration for motif scaffolding (Figure 4, Appendix F.6). Compared to leading motif scaffolding methods including RFDiffusion, Chroma and FrameFlow-Amortized, Genie 2 is capable of solving more motif scaffolding problems with more diversity (closest competitor is FrameFlow-Amortized which solves 22 problems with a total of 1,099 unique solutions vs. Genie 2 which solves 23 problems with a total of 1,445 unique solutions).
> > > - Genie 2 also provides additional supports for multi-motif scaffolding (with detailed analysis on Appendix G.3) by implicitly reasoning over the relative positions and orientations among motifs, which is a capability not supported by leading protein diffusion models such as RFDiffusion. While one can combine RFDiffusion with randomly sampled relative positions and orientations between motifs, the success rate and thus sampling efficiency is much lower than Genie 2 (Appendix G.4).
> > > - Lastly, Genie 2 is the first protein design model to successfully train on the scale of the known protein structural universe. Combined with low scale sampling (Appendix E.3), Genie 2 is able to leverage the diversity of the AlphaFold database while maintaining high designability during generation.

---

### Official Review · Reviewer_zcU3 · 2024-11-04

**Soundness:** 3
**Presentation:** 3
**Contribution:** 2
**Rating:** 5
**Confidence:** 4

**Summary:**

The paper describes Genie 2, an extension of the protein structure diffusion model Genie for motif-conditioned generation. The method is extensively benchmarked against alternative methods for unconditional and motif-conditioned diffusion models, and seems perform comparatively well.

**Strengths:**

The authors detail an extension of the Genie model for motif-conditioned protein structure generation. The incorporation of motif information is clearly explained and quite simple, and importantly seems to work quite well empirically.

Comparisons to existing methods for unconditional and motif-conditioned generation are clearly presented and fairly thorough. There are many protein structure diffusion models and the authors have done a nice job presenting results for some of the most prominent.

**Weaknesses:**

The authors highlight functional protein design as a motivation for this work, and assert that functional protein design tasks can often be reduced to scaffolding of known motifs. While this may be true for simple enzymes, many functional proteins of therapeutic and industrial interest are considerably more complex (e.g., involving conformational changes or inter-domain coordination). Without experimental validation of the results presented here, it is difficult to know whether the proposed method would meaningfully improve functional design outcomes.

Given the relationship between this work and the original Genie paper, it is surprising to not see any direct comparisons between Genie 2 and the original model. For instance, how would the original Genie architecture trained on the PDB compare with the results presented in Table 1? It seems that the largest improvements came from expanding the training dataset to include AFDB, but it would be useful to see more detailed ablations of the architectural, data, and training objective changes, and the interplay between these (if there exists any).

The analysis of low-temperature sampling is quite hard to follow as currently presented. It is not clear that the authors have used low-temperature sampling for the results presented in Table 1 until the following section analyzing secondary structure distributions. Further, in this section (line 323), the authors cite a finding that low-temperature sampling yielded better results, but do not provide the results of this analysis.

The result about “designability” of AFDB structures is interesting, but not very convincing in its current form. Rather than summarizing the results of changing from scRMSD to scTM, the authors should present the full distribution of values for both of these metrics. The authors note that ProteinMPNN may be biased towards the PDB; this hypothesis could be tested by using ESM-IF1, which has been trained on AlphaFold2-predicted structures.

Much of the evaluation presented in this paper depends on ESMFold as an oracle for structure prediction. While this has become a standard practice, as noted by the authors, it would be useful for readers to include some discussion of the shortcomings and potential biases introduced by this approach.

**Questions:**

At a couple places in the paper (line 806, line 872), the authors note that certain (earlier) model checkpoints were selected for evaluation due to better performance. How were these checkpoints selected?

It seems that the Chroma conditioner framework should admit single- and multi-motif scaffolding as described in this paper. Was there a reason this model was not benchmarked in these scenarios?

How important is the sequence information for motif conditioning? In the absence of this information, would the model still produce viable solutions to the motif scaffolding problems?

---

> ### Author Response · Authors · 2024-11-23
> **Response by Authors**
>
> We would like to thank the reviewer for the constructive and thoughtful feedback. Please see our responses below:
>
> **Weakness 1**: We agree with the reviewer that functional protein design extends beyond motif scaffolding, although the design of proteins with multiple conformations remains extremely challenging and anecdotal. We also agree that the ultimate determination of successful protein design is through experimental validation. However, given our focus in this manuscript on the core computational methodology and given the machine learning focus of ICLR, we consider experimental validation to be out of scope, and this has been the standard for protein design papers at machine learning conferences. We do believe that our in silico evaluation pipeline, which follows the norms and standards of the field as a whole, can point to meaningful improvements in design capabilities, because of the use of independent protein structure prediction methods to assess the self-consistency of designs. Furthermore, even in an experimental setting, in silico evaluation can help reduce the number of potential targets and boost experimental success rates. This was recently demonstrated in binder design [1], where in silico evaluation considerably improved success rates.
>
> **Weakness 2**: Thanks for bringing this oversight to our attention. We now provide a comparison between Genie and Genie 2 in Appendix E.1, demonstrating significant improvements from Genie to Genie 2. While we agree that it is unfair to compare Genie and Genie 2 due to differences in training dataset, model architecture, and model complexity, given limited computational resources, we decided to prioritize evaluating important hyperparameter choices in Genie 2 (e.g., number of diffusion steps and sampling noise scale) over retraining Genie in a controlled setting. This allows us to better understand various potential tradeoffs in the Genie 2 architecture, as well as explore the full capabilities of Genie 2. A full ablation study between Genie and Genie 2 would have consumed all our available computing resources and precluded our exploration of the hyperparameter space of Genie 2.
>
> **Weakness 3**: We now provide a detailed analysis on the effect of different sampling noise scales in Appendix E.3.
>
> **Weakness 4 & 5**: Thanks for this suggestion as it has led us to some new insights. We now provide further ablation studies and discussions on the self-consistency pipeline in Appendix D that we hope address all the reviewer’s concerns.
>
> **Question 1**: We provide further results and analysis on model performance across different training checkpoints in Appendix E.4, where we discuss the tradeoff in checkpoint selection. In general for checkpoint selection, we assess the 30-epoch, 40-epoch, and 50-epoch checkpointed models on unconditional generation and motif scaffolding tasks and select the best performing checkpoint. It is worth noting that the performance differences between these three checkpoints are minimal.
>
> **Question 2**: We now compare to Chroma on single-motif scaffolding in Appendix F.6. Genie 2 significantly outperforms Chroma on single-motif scaffolding, solving more problems while generating more diverse solutions at the same time. In addition, we attempt to use Chroma for multi-motif scaffolding by composing multiple substructure conditioners. However, Chroma fails to solve any multi-motif scaffolding problem.
>
> **Question 3**: We now provide an additional ablation study for single-motif scaffolding when not using motif sequence information as input in Appendix F.7. Since the functionality of a motif generally relies on side chain conformations and thus is dependent on motif sequence, we maintain our motif scaffolding settings, which is to design a protein containing matching sequence and structure to the target motif. Hence, we omit only motif sequence from the inputs to Genie 2 but maintain the motif residue identities during inverse folding (when using ProteinMPNN). Under these settings, Genie 2 remains capable of providing viable solutions for the majority of motif scaffolding problems, but we do observe a drop in its performance, suggesting that Genie 2 is internalizing sequence information into its latent representations to improve protein design.
>
> **Reference**
>
> [1] Bennett, Nathaniel R., Brian Coventry, Inna Goreshnik, Buwei Huang, Aza Allen, Dionne Vafeados, Ying Po Peng et al. "Improving de novo protein binder design with deep learning." Nature Communications 14, no. 1 (2023): 2625.

---

> ### Author Response · Authors · 2024-11-29
>
> We appreciate the time and effort you have put in reviewing our work. As the discussion period is coming to close (Dec. 2nd), we would like to ask if our replies and added results have helped to address your questions and concerns and if there is any further clarification we could provide.

---

### Official Review · Reviewer_NPXG · 2024-11-07

**Soundness:** 2
**Presentation:** 2
**Contribution:** 2
**Rating:** 5
**Confidence:** 5

**Summary:**

The authors introduce Genie 2, an extension of Genie that incorporates architectural innovations and extensive data augmentation. Genie 2 supports both unconditional generation and multi-motif scaffolding. The authors claim that Genie 2 achieves state-of-the-art performance, outperforming all known methods on key design metrics.

**Strengths:**

1. For unconditional generation, Genie 2 performs better than Genie and other methods when generating short proteins.
2.  Genie 2 also supports multi-motif scaffolding.

**Weaknesses:**

1. The authors overstate the performance of Genie 2 in the Abstract. While they claim that Genie 2 achieves state-of-the-art performance and outperforms all known methods on key design metrics, Figure 3 Panel B shows that Genie 2 performs significantly worse than Proteus on long proteins.

2. The comparison benchmarking is insufficient.

    a. Unconditional generation: The comparison lacks the inclusion of recent methods such as MultiFlow [1], CarbonNovo [2], and FoldFlow2 [3], which have demonstrated significant performance improvements.

    b. Single-Motif Scaffolding: Methods like Chroma and FoldFlow2 should be included for comparison. While Chroma is included in the benchmarking for unconditional generation, it is excluded in the scaffolding generation comparisons. Clarification on this omission is needed.
3. The methodological contributions appear incremental. The authors extend Genie's architecture and training procedure to enable motif scaffolding, with most modifications focusing on adapting input features. Additionally, Genie 2 incorporates triangular multiplicative updates, which have already been shown to be effective in previous works such as Proteus [4], CarbonNovo [2], and FoldFlow2 [3].
4. Insufficient Analysis of Multi-Motif Scaffolding: Despite claiming multi-motif scaffolding as a key contribution, the paper provides limited analysis in this area. Interesting analysis would be :

    a. In failed cases, is the failure due to incorrect single motif design or inaccuracies in the relative positioning of motifs?

    b. Does the success rate of scaffolding depend on the number of motifs being used?

 5.  The method requires significantly more inference steps than previous methods, indicating low sampling efficiency.


[1] Campbell et al. “Generative Flows on Discrete State-Spaces: Enabling Multimodal Flows with Applications to Protein Co-Design”. ICML 2024. https://proceedings.mlr.press/v235/campbell24a.html

[2] Ren et al. “CarbonNovo: Joint Design of Protein Structure and Sequence Using a Unified Energy-based Model”. ICML 2024. https://proceedings.mlr.press/v235/ren24e.html

[3] Huguet et al. “Sequence-Augmented SE(3)-Flow Matching for Conditional Protein Backbone Generation”. https://arxiv.org/abs/2405.20313

[4] Wang et al. “Proteus: Exploring Protein Structure Generation for Enhanced Designability and Efficiency”. ICML 2024. https://proceedings.mlr.press/v235/wang24bi.html

**Questions:**

1. Previous methods tend to generate more alpha-helices than beta-strands. In Genie 2, what is the distribution of generated proteins among the structural classes of all-alpha, all-beta, alpha+beta, and alpha/beta? Does the designability of the generated proteins depend on the ratio of alpha-helices?

2. Why was a batch size of 256 chosen for training Genie 2? Both Proteus and Chroma use batch sizes larger than 500, yet they exhibit significantly different performance levels. Therefore, without additional experimental results, it is unclear whether batch size is the main reason for Genie 2's significantly worse performance compared to Proteus on long proteins.

2. Please address the concerns and questions raised in the Weaknesses section.

---

> ### Author Response · Authors · 2024-11-23
> **Response by Authors (Part 1/2)**
>
> We would like to thank the reviewer for the constructive and thoughtful feedback. Please see our responses below:
>
> **Weakness 1**: As mentioned in Section 4.1, designability could be sometimes misleading since it does not account for structural diversity. Relating to Figure 3B, while Proteus achieves higher designability than Genie 2 on longer proteins, its diversity is similar or even lower than Genie 2 as shown in Figure 3C. This suggests that Proteus repeatedly generates the same or highly similar structures.
>
> Since our initial submission, we trained Genie 2 for additional epochs, which has led to improved performance, particularly on longer proteins. We include further results and analysis in Appendix E.4. Genie 2 now achieves superior performance over all competing methods in generating diverse long proteins. Thus, we believe that Genie 2 outperforms Proteus in generative performance, even in the generation of long proteins.
>
> Nonetheless, we will revise our wording in the manuscript to more precisely describe how Genie 2 outperforms existing methods.
>
> **Weakness 2**: For unconditional generation, we now provide further comparisons with MultiFlow and CarbonNovo in Appendix E.7. The code for FoldFlow-2 is unavailable, precluding direct comparison with Genie 2. Compared to MultiFlow and CarbonNovo, Genie 2 achieves comparable or better designability while significantly outperforming them on diversity and novelty. For single-motif scaffolding, we additionally compare to Chroma and provide the results and analysis in Appendix F.6. Genie 2 significantly outperforms Chroma on motif scaffolding, solving more problems while generating more diverse solutions at the same time.
>
> **Weakness 3**: We respectfully disagree on this point. While our methodological advances may seem incremental, in the sense that we use known components such as triangular multiplicative updates employed by other methods such as Proteus and FoldFlow2, our architecture is novel, i.e., how we use and combine those components is different. In particular, the Genie 2 architecture is the current state-of-the-art against all methods that have been suggested by the reviewers and that we previously compared against, on the key design metrics of designability, novelty, and diversity. We believe that achieving SOTA results merits consideration.
>
> [We add the following additional point, copied from our response to Reviewer 4r8p, regarding the additional significance and originality of training on the AFDB database] We believe that our improved diversity results from the combined effect of training on AFDB and sampling at a lower noise scale. Training on AFDB allows us to leverage a much larger structural space than that of the PDB and results in a better protein diffusion model that can tackle new protein design tasks unaddressed by existing methods. Training a leading protein diffusion model on the whole of the clustered AFDB database, while apparently conceptually simple, in fact requires overcoming substantial engineering challenges as well as making critical design choices. This is supported by the fact that while AFDB has been available since 2021, Genie 2 is the first to learn to model the data distribution of AFDB structures. Some challenges of modeling the data distribution of AFDB structures include structural quality (since they are not experimentally validated), existence of multi-domain structures and the scale of AFDB (over 200 millions structures). Indeed, soon after Genie 2 was preprinted, FoldFlow-2 came out as a contemporary method that attempted to use AFDB, but found that training on AFDB actually  hampered model performance. An innovation of this work is the use of low noise-scale sampling in combination with AFDB, which we believe contracts average protein radius and encourages the generation of more compact (and thus more designable) structures (refer to Appendix E.3 for more discussion on low sampling noise scale). By combining training on AFDB with low noise-scale sampling, Genie 2 is able to leverage the structural diversity of AFDB while retaining high quality generation capability, enabling successful utilization of AFDB.

---

> ### Author Response · Authors · 2024-11-23
> **Response by Authors (Part 2/2)**
>
> **Weakness 4**: Thank you for your suggestions and we now provide further case studies on the two multi-motif scaffolding problems that Genie 2 fails on in Appendix G.3. In addition, we provide a controlled experiment on scaffolding an increasing number of EF-hand $Ca^{2+}$-binding motifs in Appendix G.3.3. As the number of motifs increases, multi-motif scaffolding performance drops since it is hard to find a viable solution that satisfies all motif constraints; however, as the number of motifs increases, the number of unique successes increases (bounded by the number of successes), possibly because the increase in sequence length provides a larger search space for solutions. Furthermore, we now compare Genie 2’s performance on multi-motif scaffolding with RFDiffusion in Appendix G.4, demonstrating that Genie 2 has an edge on multi-motif scaffolding.
>
> **Weakness 5**: We now provide further ablation studies on the number of diffusion steps in Appendix B.2. As shown in Table 4, while using 1,000 diffusion steps does achieve the best performance on both unconditional generation and motif scaffolding, Genie 2’s performance with fewer diffusion steps is comparable. Genie 2 thus allows for fast sampling with minimal loss of generative capabilities.
>
> **Question 1**: Using the set of Genie 2-generated structures (curated for analysis in Table 1), we have now visualized the distribution of scRMSD as a function of the percentage of helices (Appendix E.6). We observe that while scRMSD slightly decreases with higher helical fraction, the majority of the generated structures have relatively low scRMSD regardless of secondary structure composition. This, combined with the secondary structure distribution shown in Figure 2A, suggests that while Genie 2 has a tendency to generate more helical structures, its generation quality of either helical or stranded structures is similar.
>
> **Question 2**: We believe the reviewer meant maximum sequence length instead of batch size for this question, but please correct us if we are mistaken. We set the maximum sequence length to 256 due to computational limitations. As Genie 2 utilizes triangular multiplicative update layers, it has $O(N^3)$ space complexity where N is the maximum sequence length. However, despite being trained only on proteins with up to 256 residues, Genie 2 is capable of generating proteins beyond 256 residues in length. As discussed in Weakness 1, compared to existing methods, Genie 2 is capable of generating more diverse structures, even for longer proteins. We would like to also note that for practical applications, the majority of protein design applications focus on proteins shorter than 200 residues.

---

> > ### Comment · Reviewer_NPXG · 2024-11-25
> > **thank you for your response**
> >
> > Thank you for your response. I have increased my score in recognition of your hard work in the rebuttal, although my main concerns remain. Good luck.
> >
> > Here is an additional point: Chroma also leverages low noise-scale sampling to trade diversity for quality, just as you trade diversity for designability. They refer to this technique as low-temperature sampling in their paper (Appendix C) and provide a comprehensive discussion of its pros and cons. Thus, I suggest that you cite Chroma when you mention this technique in your paper.

---

> > > ### Author Response · Authors · 2024-11-29
> > > **Further clarification on low noise scale sampling**
> > >
> > > As mentioned in Appendix E.3, we would like to highlight a distinction between low temperature sampling (used by Chroma) and low scale sampling (used by Genie 2). For low temperature sampling, Chroma uses annealed Langervin dynamics to achieve approximate equilibrium (or in other words, reweighing populations according to the temperature-annealed distribution). In our case, using low scale sampling is a heuristic that we use to contract the protein radius in the generation process (in an attempt to reduce the flexibility inherit in the AFDB dataset). Thus, even though low temperature sampling and low scale sampling achieve similar results (lower diversity for higher designability), the motivations are different. We would cite Chroma in Appendix E.3 and highlight this difference more clearly for our revised version.

---

> > > ### Author Response · Authors · 2024-11-29
> > > **Thank you for your response**
> > >
> > > Thank you for your response and increasing your score to 5. We appreciate the time and effort you have put in reviewing our work. We would like to ask if it is possible to provide further clarification for how the manuscript could have been improved above a score of 5 (below acceptance threshold). Here we note that Genie 2 is the current SOTA method for unconditional generation and motif scaffolding. In particular, in the revised version of our manuscript we have demonstrated that:
> > >
> > > - Genie 2 outperforms all methods in consideration including RFDiffusion, Chroma, CarbonNovo and MultiFlow by achieving comparable designability (0.96) but higher diversity and novelty (Appendix E.7; closest competitor is RFDiffusion which achieves F1: 0.76, diversity: 0.63, and novelty: 0.26 vs. Genie 2 which achieves F1: 0.93, diversity: 0.91, and novelty: 0.41). We note that the deltas are rather substantial, and so this is not merely an incremental claim of achieving SOTA.
> > > - These results are achieved even with much fewer diffusion steps and thus higher sampling efficiency (Appendix B.2, H). Together these results establish a much stronger foundation than what is currently available for future protein design methods.
> > > - Genie 2 significantly outperforms existing methods on generating longer proteins (Appendix E.4), generating significantly more diverse designable clusters than existing methods for proteins with more than 300 residues (2.5-fold higher diversity compared to Proteus on generating proteins with 500 residues)
> > > - Genie 2 outperforms all methods in consideration for motif scaffolding (Figure 4, Appendix F.6). Compared to leading motif scaffolding methods including RFDiffusion, Chroma and FrameFlow-Amortized, Genie 2 is capable of solving more motif scaffolding problems with more diversity (closest competitor is FrameFlow-Amortized which solves 22 problems with a total of 1,099 unique solutions vs. Genie 2 which solves 23 problems with a total of 1,445 unique solutions).
> > > - Genie 2 also provides additional supports for multi-motif scaffolding (with detailed analysis on Appendix G.3) by implicitly reasoning over the relative positions and orientations among motifs, which is a capability not supported by leading protein diffusion models such as RFDiffusion. While one can combine RFDiffusion with randomly sampled relative positions and orientations between motifs, the success rate and thus sampling efficiency is much lower than Genie 2 (Appendix G.4).
> > > - Lastly, Genie 2 is the first protein design model to successfully train on the scale of the known protein structural universe. Combined with low scale sampling (Appendix E.3), Genie 2 is able to leverage the diversity of the AlphaFold database while maintaining high designability during generation.

---

> ### Comment · Reviewer_NPXG · 2024-12-02
> **My main concerns remain**
>
> Thank you for your further summary. However, my main concerns remain.
>
> 1. **The method's novelty seems very incremental**
>
>     The proposed method appears to be a simple combination of existing components **that have already been proven useful in backbone generation**, and thus it does not offer significant novelty. Specifically:
>
>     a. **Model architecture**: Genie2's denoiser network mainly includes an Evoformer-like architecture and an Invariant Point Attention (IPA) module. It shares a very similar architecture with CarbonNovo's structure module (see Figure 1 and Algorithm 1 in their paper). Proteus also incorporates an IPA and an Evoformer-like architecture called the Graph Triangle Block (Figure 2 in their paper). FoldFlow++ also leverages a similar architecture (Figure 1 in their paper). I also note that these previous works have been peer-reviewed in ICML 2024 or NeurIPS 2024, not just in their pre-print versions.
>
>
>     b. **Low noise scale sampling**:  Your clarification in your response is not accurate.  Low noise scale sampling and low-temperature sampling essentially have the same meaning, both in motivation and technique. In both cases, the motivation is to trade diversity for quality. In your method, quality is measured in terms of designability.
>
>     c. **Utilizing the AlphaFold Database (AFDB)**: Training the model by combining existing techniques that have been proven useful for backbone generation on more data (from AFDB) seems like a very trivial idea to achieve SOTA.
>
>
> 2. **Concern on overstatement remains**
>
>     The claim of "architectural innovations" in the abstract is very questionable, as your method shares a very similar architecture with AlphaFold2. Moreover, these architectural components **have already been proven effective in backbone generation**, as stated in my comment 1.a.

---

### Author Response · Authors · 2024-11-23

We would like to thank all reviewers for their careful reviews and constructive feedback. We hope that our response will clarify any doubts or concerns. For the rebuttal, we kept the bulk of our updates in the Appendix, with changes highlighted in red for clarity. We will integrate these additional results and discussions into the main text for our final revised version of the manuscript. Our key additions include
- Evaluation results on a version of Genie 2 trained for additional epochs, demonstrating that when trained for longer, Genie 2 generates significantly more diverse long proteins (>300 residues) compared to existing methods (Appendix E.4)
- Evaluation results on a version of Genie 2 trained with fewer diffusion steps, demonstrating that Genie 2 can achieve comparable generative performance even with fewer diffusion steps (Appendix B.2) and has comparable or much higher sampling efficiency compared to existing methods (Appendix H).
- Comparisons with MultiFlow, CarbonNovo, and Chroma, which demonstrate that Genie 2 achieves better performance on both unconditional generation and motif scaffolding relative to all other methods (Appendix E.7 and F.6).
- Additional analysis on multi-motif scaffolding performance, including comparison with RFDiffusion, analysis of failed problems, and a controlled experiment studying the effect of number of motifs on scaffolding performance (Appendix G.3 and G.4).
- A discussion of the strengths and limitations of the self-consistency pipeline, with further analysis on the discrepancy in self-consistency evaluations between PDB and AFDB structures (Appendix D).
- A discussion on the effect of sampling with lower noise scale, which allows Genie 2 to leverage the structural diversity of AFDB while retaining high designability during the sampling process (Appendix E.3).

---

### Meta-Review · Area_Chair_YxH6 · 2024-12-21

**Metareview:**

This paper introduces Genie 2, which extends the Genie's capabilities through architectural innovations and extensive data augmentation, enabling it to explore a broader and more diverse protein structure space. Genie 2 achieves state-of-the-art performance in both unconditional and conditional protein generation, surpassing existing methods on metrics like designability, diversity, and novelty. It also excels in solving motif scaffolding problems with more unique and varied solutions, setting a new benchmark for structure-based protein design. The reviewers expressed concerns regarding the novelty and significance of the proposed method. For example, the claimed architectural innovation and the data augmentation from AFDB have been widely employed in many other works, and the method underperforms compared to Proteus in designing long protein structures (>300 residues). Despite the authors' rebuttal and additional experiments, the paper did not gain sufficient support. Therefore, I recommend rejection.

**Additional Comments On Reviewer Discussion:**

The reviewers raised concerns about the novelty and significance of the proposed method. Specifically: (1) the claimed architectural innovation and the data augmentation from AFDB have been widely employed in many other works, (2) Genie 2 underperforms Proteus on long protein structure design (>300 residues), and (3) the paper lacks important baselines, such as Multiflow and CARBONNOVO, for unconditional structure generation. During the rebuttal, the authors added numerous experiments including additional baselines, which led some reviewers (NPXG,pv9F,4r8p) to increase their scores to 5. However, the paper still failed to gain sufficient support. Therefore, I recommend rejection.

---

### Decision · Program_Chairs · 2025-01-22

Reject